# Rhizobacteria opportunistically boost colonization and impair plant fitness by degrading plant-derived coumarins under iron deficiency

Yichao Gu[1], Piaopiao Pan [1], Gang Yu [2] & Ning-Yi Zhou [1]✉

Plants recruit root-associated bacterial assemblies primarily through the secretion of specialized metabolites, and the resultant rhizospheric microbiota is empirically considered beneficial. However, detrimental effects on plants arising from bacterial colonization that exploits plant-derived metabolites are rarely documented. Here, we demonstrate that the rhizosphere-derived *Pseudomonas* sp. strain NyZ480 exhibits a versatile capacity to effectively degrade and utilize simple coumarins—a class of root exudates essential for plant iron acquisition and pathogen defense. This robust catabolic capability is mediated by conserved genetic determinants in NyZ480. In particular, redundant degradation-initiating *xenA* genes confer NyZ480 not only growth using simple coumarins but also resistance to these antimicrobial metabolites. Consequently, NyZ480 significantly colonizes iron-stressed, coumarin-secreting Arabidopsis roots, trapping plants in perpetual iron scarcity and progressively compromising iron acquisition and overall fitness. Bioinformatic analyses indicate that *xenA* homologs are prevalent and redundant in environmental bacteria. Thus, we reveal a rhizospheric phenomenon where microorganisms opportunistically utilize and detoxify host-secreted specialized metabolites under stress conditions, enhancing colonization and impairing plant fitness.

Plant metabolites are low-molecular-mass natural products crucial for plant growth and development[1]. Unlike essential primary metabolites, specialized metabolites (hereafter, SMs) are conditionally produced by plants either for compatibility with the surroundings or serve as defense mechanisms against pathogen invasions[2]. Among SMs, simple coumarins, biosynthesized via the phenylpropanoid pathway[3], have gained increasing attention over the past decade as a notable class of plant natural products[4–6]. Genetic determinants involved in simple coumarin synthesis are widely distributed across the plant kingdom[7], highlighting the functional importance of these phytocompounds.

Similar to other SMs, simple coumarins, including esculetin, fraxetin, scopoletin, and sideretin, are synthesized by plants upon elicitation of external stresses[8–10]. For instance, esculetin was identified as an Fe(III)-chelating coumarin released by *Arabidopsis thaliana* (hereafter, Arabidopsis) under iron scarcity[11]. Fraxetin, another key coumarin, has been determined to simultaneously chelate and reduce Fe(III), thereby facilitating iron uptake under neutral or alkaline

[1]State Key Laboratory of Microbial Metabolism, Joint International Research Laboratory of Metabolic and Developmental Sciences, and School of Life Sciences and Biotechnology, Shanghai Jiao Tong University, Shanghai, China. [2]Shanghai Collaborative Innovation Center of Agri-Seeds, Joint Center for Single Cell Biology, School of Agriculture and Biology, Shanghai Jiao Tong University, Shanghai, China. ✉e-mail: ningyi.zhou@sjtu.edu.cn

conditions[12,13]. Sideretin, derived from fraxetin, is a redox-active catecholic coumarin exuded by Arabidopsis roots in response to iron deprivation under acidic conditions[7]. Collectively, simple coumarins play pivotal roles in plants' acquisition of Fe(III), a less bioavailable form of iron.

Not only are simple coumarins produced under abiotic pressure, but they also serve as key mediators in biotic interactions. For example, scopoletin, an antimicrobial coumarin, is prominently produced and exuded by Arabidopsis roots upon the onset of rhizobacteria-mediated induced systemic resistance (hereafter, ISR) as well as under iron deficiency[14]. Upon its secretion into the rhizosphere, scopoletin alters the microbial community assembly[15], thereby resulting in a plant-driven selection of root microbiota that promotes host fitness. On the one hand, this ISR-elicited exudation enhances plant defense against sensitive soil-borne fungal pathogens[14,16]; on the other hand, it enriches coumarin-tolerant microbes that may exert beneficial functions, such as producing siderophores to alleviate plant iron limitation[17]. While the exact composition and quantity of simple coumarins exuded by pathogen-infected plants remain to be fully characterized, transcriptomic analyzes have preliminarily indicated that foliar infection activates biosynthesis of simple coumarins[18]. Moreover, experimental evidence indicates that several simple coumarins, including coumarin, daphnetin, esculetin, scopoletin, and umbelliferone, exhibit antimicrobial activity[19]. Together, these findings support the potential dual functions of simple coumarins in plants subjected to biotic and abiotic stresses.

Since SMs are secreted into the rhizosphere as root exudates, they likely mediate sophisticated interactions between plants and rhizobacteria. Typically, the dual nature of these phytocompounds is reflected in both the recruitment of microbial members that utilize or detoxify the corresponding substrates[20] and the suppression of those that are sensitive to their presence[21,22]. Hence, plant-derived molecules like simple coumarins are believed to strongly sculpt the rhizomicrobiome in ways that benefit plant health[21,23]. However, such plant-mediated selection can be surprisingly subverted by adaptable pathogens. For instance, *Ralstonia solanacearum* can degrade the plant defense hormone salicylic acid to enhance its virulence on tobacco[24]; *Phytophthora sojae* even proactively boosts the biosynthesis of trehalose in soybean and exploits it as a carbon source to support infection[25]. Hence, the associations between major plant SMs and the related microorganisms should be clearly deciphered, so as to effectively manage both probiotic and pathogenic phytobacteria.

The significant roles of simple coumarins in plant iron acquisition and pathogen defense have already been extensively studied at the physiological and genetic levels[7,9,11,12,14,26–32]. However, little effort has been devoted to elucidating how plant-associated microbes respond to these phytocompounds and how these interactions consequently affect plant fitness. In our previous study, a coumarin degrader *Pseudomonas* sp. strain NyZ480[33,34] (hereafter, NyZ480) was characterized from the rhizosphere of *Plantago asiatica* L., and it was employed here to investigate the effects of microbial degradation of simple coumarins on plants' welfare. Our findings establish that bacterial utilization and detoxification of plant-derived simple coumarins drive enhanced bacterial colonization and worsened growth impairment in iron-stressed Arabidopsis. The widespread occurrence and high redundancy of the degradation-initiating gene *xenA* among diverse bacterial genomes (43,013/51,445 genomes) suggest a massive biodegradation of simple coumarins in the environment. Given that simple coumarins are widely distributed across the plant kingdom, a potential evolutionary "arms race" between plants and rhizobacteria can be inferred. Together, this work demonstrates an unreported rhizosphere paradigm where bacterial exploitation of plant SMs under iron deficiency leads to opportunistic bacterial colonization and severe detriment to the host.

## Results

### Simple coumarins produced by iron-deficient Arabidopsis are degraded and utilized by NyZ480

Iron is an essential micronutrient for plant growth, as it plays vital roles in various cellular processes, including mitochondrial respiration, repair of nucleotides, and chlorophyll biosynthesis[35]. The growth phenotype of Arabidopsis varies with iron availability. Here, iron deficiency-induced chlorosis was more obviously observed in the shoots of Arabidopsis Col-0 (hereafter, Col-0) when plants were supplied with deficient and less bioavailable iron (the canonical 100 μM $Fe^{2+}$ in ½ Murashige Skoog (½ MS) media was replaced by 10 μM $Fe^{3+}$, Fig. 1a). Since simple coumarins have previously been reported to be indispensable for plants under iron scarcity[8,10,36], their secretion was preliminarily assessed according to the root autofluorescence under UV light. Results showed that Col-0 roots grown on iron-limited (10 μM $Fe^{2+}$ or 10 μM $Fe^{3+}$) ½ MS agar plates emitted stronger average fluorescence intensities than those grown under iron-sufficient conditions (100 μM $Fe^{2+}$ or 100 μM $Fe^{3+}$, Fig. 1a). A similar pattern was observed for hydroponic Col-0 plants (Supplementary Fig. 1). Quantitative analysis of exuded simple coumarins in the spent hydroponic media showed that the concentrations of coumarin, fraxetin, and scopoletin increased as iron supply decreased, and that exudation was more strongly stimulated by the less bioavailable $Fe^{3+}$. Among these, scopoletin was the most abundantly secreted simple coumarin under iron deficiency, reaching a concentration of up to nearly 4000 ng/mL (Fig. 1b). Collectively, these observations confirmed that iron-stressed Col-0 plants enhanced the synthesis and exudation of simple coumarins as a responsive strategy. However, upon inoculation with the previously obtained coumarin degrader NyZ480[33,34], none of these exudates were detected (Fig. 1b).

As coumarin, esculetin, fraxetin, and scopoletin share the same core structure, it was proposed that NyZ480 could efficiently degrade these iron deficiency-induced phytocompounds. Subsequent biodegradation assays confirmed that NyZ480 exhibited degradation of all four simple coumarins (Fig. 1c). Within 6 h, 0.5 mM concentration of coumarin, esculetin, and fraxetin were completely eliminated despite the slight spontaneous degradation of fraxetin due to its inherent instability (Supplementary Fig. 2). Although NyZ480 was isolated with coumarin as its sole carbon source[33], fraxetin was also found to be utilized for its growth (Fig. 1d). It was further hypothesized that the catabolism of both coumarin and fraxetin in NyZ480 may be achieved via the same genetic determinants.

RNA-seq analysis was employed to profile the gene expression changes of NyZ480 in response to the presence of a simple coumarin mixture (group EFS). Compared to the control group (treated with glucose, group GLU), a total of 735 genes were up-regulated ($p < 0.05$, $Log_2FC > 1$) while 832 genes were down-regulated ($p < 0.05$, $Log_2FC < -1$) (Supplementary Fig. 3a). Bacterial mobility of NyZ480 was likely affected by these phytocompounds, as the differentially expressed genes (DEGs) were enriched in Kyoto Encyclopedia of Genes and Genomes (KEGG) pathways of "flagellar assembly" and "bacterial chemotaxis" (Supplementary Fig. 3b), which resembled the responsive pattern of NyZ480 to coumarin alone[33]. Moreover, a number of simple coumarin-induced DEGs showed enrichment in GO terms associated with the "organic substance metabolic process" (Supplementary Fig. 3c), indicating the catabolic potential of NyZ480 towards these substrates. Since the catabolic genes of coumarin had already been identified in our previous study[33], their transcription levels were specifically investigated after simple coumarin induction. Significant up-regulation of three dispersed *xenA* homologs (the degradation-initiating gene: *xenA1*, *xenA3*, and *xenA7*), of *couC*, and of a partial *mhp* cluster (the downstream catabolic genes: *mhpB*, *mhpC*, *mhpD*, and *mhpT*) was recorded (Fig. 1e; group EFS; $p < 0.05$, $Log_2FC > 1$), suggesting that this specific gene set modulates the degradation dynamics of coumarin compounds.

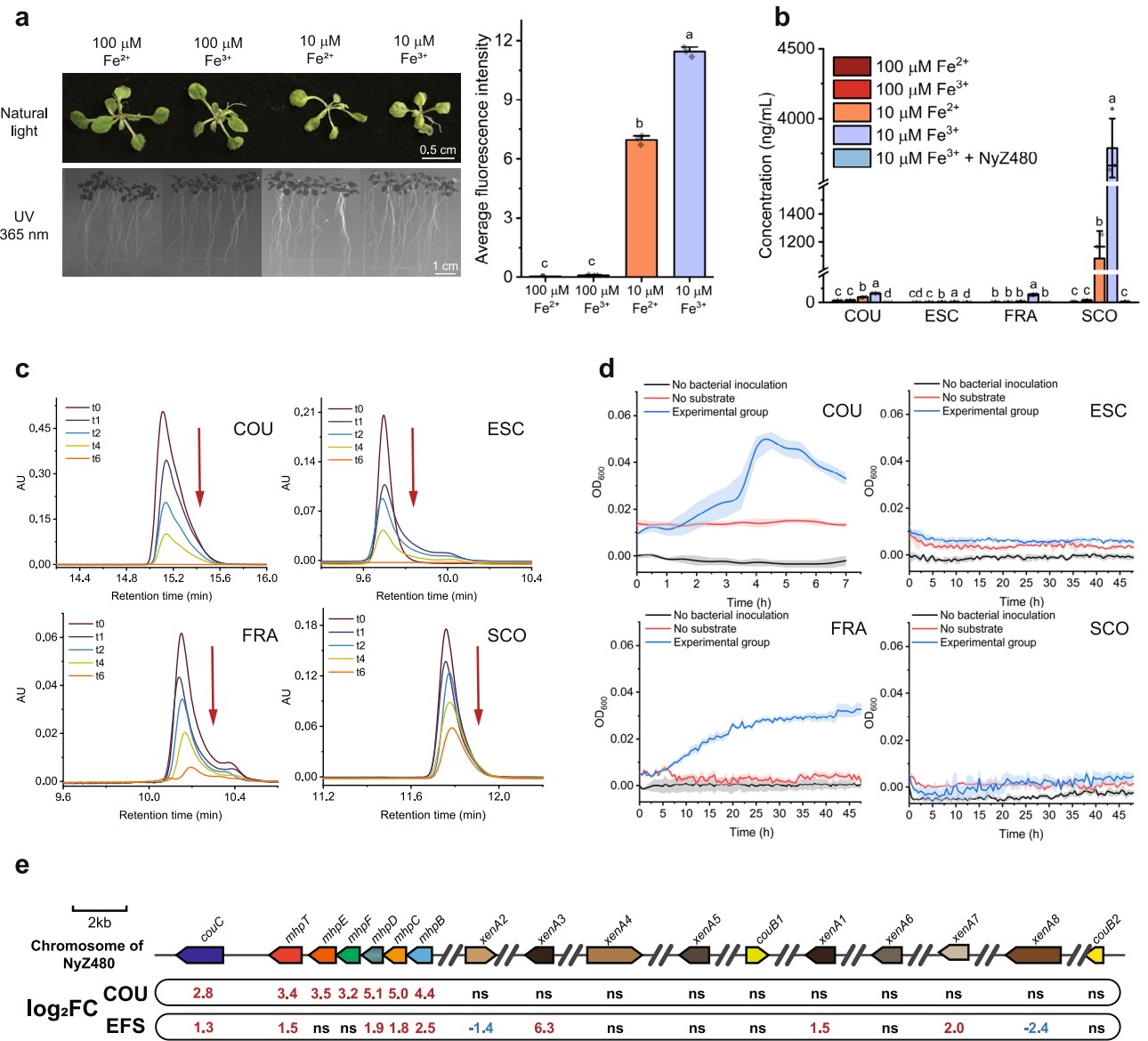

**Fig. 1 | NyZ480 degrades and utilizes simple coumarins exuded by iron-limited Arabidopsis Col-0. a** Quantification of average fluorescence intensity of the differently cultured Col-0 roots. Plants were grown on ½ Murashige Skoog (½ MS) agar plates containing 100 μM $Fe^{2+}$, 100 μM $Fe^{3+}$, 10 μM $Fe^{2+}$ or 10 μM $Fe^{3+}$ for two weeks. Representative images were taken under bright and UV light (365 nm). Data are mean values of three biological replicates ($n = 3$). Each biological replicate consisted of one plate containing six seedlings. **b** Quantification of simple coumarins including coumarin (COU), esculetin (ESC), fraxetin (FRA) and scopoletin (SCO) in the spent liquid ½ MS media of 18-day-old Col-0 by liquid chromatograph mass spectrometer (LC-MS). NyZ480 inoculation was performed on 16-day-old Col-0 plants that had been pre-grown under 10 μM $Fe^{3+}$, a condition under which simple coumarin exudation is maximized. **c** Degradation of COU, ESC, FRA, and SCO by NyZ480 over time. HPLC chromatograms were captured during the time course of the degradation assays (0, 1, 2, 4, 6 h), and red arrows indicate the degradation of each compound over time. **d** Growth of NyZ480 with COU, ESC, FRA, and SCO as its sole carbon sources. Solid lines represent the mean values of three biological replicates ($n = 3$), and shaded areas indicate the standard deviation (SD). Control groups without bacterial inoculation or substrate were included. **e** Gene expression was profiled in response to induction by COU alone (group COU) or a mixture of ESC, FRA, and SCO (group EFS). Two-tailed Wald test in DESeq2[55] was used to calculate $p$-values. Significantly up-regulated ($p < 0.05$, $\log_2FC > 1$), down-regulated ($p < 0.05$, $\log_2FC < -1$), and non-significant (ns) genes are highlighted in red, green, and black, respectively, with their $\log_2FC$ values displayed. Statistical differences in **a**, **b** were determined by one-way ANOVA with Tukey's post hoc test ($p < 0.05$), and the exact $p$-values are provided in the Source Data file. Error bars in **a**, **b** indicate ± SD of biological replicates.

## Redundant XenAs in NyZ480 initiate the degradation of simple coumarins

In the previously reported coumarin-induced transcriptome of NyZ480 (Fig. 1e; group COU), all eight identified *xenA* homologs (*xenA1-8*) in the bacterial genome were found constitutively expressed and not induced by coumarin alone[33]. However, among these, five *xenA* genes (*xenA1, xenA3, xenA5, xenA6, xenA7*) were experimentally confirmed to encode functional enzymes that initiate degradation[34]. Given that *xenA1, xenA3,* and *xenA7* were significantly up-regulated in

response to simple coumarins (Fig. 1e; group EFS), it was speculated that the redundant XenA homologs in NyZ480 may function cooperatively in the versatile degradation of these coumarin compounds.

Subsequent whole-cell biotransformation assays demonstrated that esculetin, fraxetin, and scopoletin were each degraded by multiple heterologously expressed XenA enzymes. Extracted ion chromatograms (EICs) showed that fraxetin (retention time: 16.7 min; [M-H]⁻, $m/z$ 207.0290) and scopoletin (retention time: 18.7 min; [M-H]⁻, $m/z$ 191.0367) were both transformed by XenA1, XenA3, XenA5, XenA6, and

XenA7, whereas six XenAs (all eight except XenA1 and XenA6) were functional against esculetin (retention time: 16.6 min; [M-H]⁻, $m/z$ 177.0208) (Fig. 2a, b, and Supplementary Fig. 4a, b). Transient intermediates generated by XenA catalysis were identified for esculetin (retention time:13.2 min; [M+H]⁺, $m/z$ 181.0280), fraxetin (retention time: 17.1 min; [M+H]⁺, $m/z$ 211.1450), and scopoletin (retention time: 15.3 min; [M+H]⁺, $m/z$ 195.0288). Due to their instability, accumulation of the final hydrolysis products of these intermediates was also captured: an esculetin-derived compound (retention time: 31.0 min; [M-H]⁻, $m/z$ 197.1547), a fraxetin-derived compound (retention time: 28.9 min; [M-H]⁻, $m/z$ 227.2017), and a scopoletin-derived compound (retention time: 16.2 min; [M-H]⁻, $m/z$ 211.0618).

These results clearly indicated that the redundant *xenA* genes play essential roles in initiating the degradation of simple coumarins. Given that these plant-derived compounds are widely present in the rhizosphere and exhibit antimicrobial activity, the redundant *xenA* genes harbored by the rhizosphere-isolated NyZ480 likely represent an adaptive detoxification strategy, evolved in the ongoing tug-of-war between rhizospheric microorganisms and their host plants.

## The enhanced colonization of NyZ480 on iron-deficient Arabidopsis Col-0 disrupts iron acquisition and impairs plant fitness

Since iron-limited Col-0 secretes simple coumarins for iron acquisition and NyZ480 can degrade and utilize these compounds, we hypothesized that this strain would proliferate on the roots of such plants under iron deficiency. To test this, wild-type NyZ480 (WT-NyZ480) was inoculated onto Col-0 roots grown under conditions of either iron-sufficiency (100 μM $Fe^{2+}$) or iron-deficiency (10 μM $Fe^{3+}$). Colony forming unit (CFU) assays indicated that WT-NyZ480 proliferated significantly more on iron-deficient roots than on iron-sufficient roots, with the $\log_{10}$(CFU/root) value reaching 7.5 at 7 days post inoculation (DPI) (Fig. 3a). Although a slight increase in bacterial biomass was observed on iron-sufficient seedlings ($\log_{10}$(CFU/root) value exceeding 4 at 7 DPI), it was negligible compared with that under iron deficiency. This enhanced colonization on iron-deficient Col-0 roots was further confirmed by confocal laser scanning microscopy (CLSM) at 7 DPI using *rfp*-tagged NyZ480 (tagged with red fluorescence protein) (Supplementary Fig. 5a). Collectively, these observations demonstrated that opportunistic colonization by NyZ480 was strongly promoted by the iron-deficient status of the host plant.

Phenotypic changes of Col-0 were characterized to assess the consequences of WT-NyZ480 colonization. Under both iron-sufficient and iron-deficient conditions, WT-NyZ480 inoculation significantly reduced plant fresh weight, primary root length, and lateral root count. A significant decrease in total chlorophyll concentration was also observed in Col-0, but only under iron-deficient conditions (Fig. 3b). Taken together, these results indicated that WT-NyZ480 colonization generally impaired Arabidopsis fitness, and that iron-deficient culture conditions particularly boosted proliferation and exacerbated its detrimental effect.

Since chlorophyll concentration is closely associated with plant iron status, it was proposed that WT-NyZ480 degrades iron-mobilizing simple coumarins, thereby hampering coumarin-facilitated iron acquisition and leading to a reduction in total chlorophyll levels. This hypothesis was supported by direct determination of plant iron concentration using inductively coupled plasma-mass spectrometry (ICP-MS) (Fig. 3c), which revealed a further decrease in the shoot iron concentration in iron-deficient Col-0 after WT-NyZ480 inoculation. This confirmed that WT-NyZ480 indeed interfered with iron absorption in plants under iron stress, and the worsened bacterial detriment to plants was iron deficiency-dependent.

## NyZ480 causes transcriptional reprogramming in Arabidopsis Col-0

To reveal the transcriptional responses of Col-0 to WT-NyZ480 colonization, RNA-seq analysis was conducted on root tissues under distinct culture conditions: 1. iron sufficiency (control); 2. iron deficiency (dFe); 3. WT-NyZ480 inoculation (NyZ480); 4. a combinatorial treatment of iron deficiency and WT-NyZ480 inoculation (dFe&NyZ480). Compared with control plants, iron deficiency had the mildest effects, with 441 DEGs (Fig. 4a; $p < 0.05$, $|\text{Log}_2\text{FC}| > 1$) identified, including 225 up-regulated and 186 down-regulated genes. Most up-regulated genes were enriched in Gene Ontology (GO) terms related to stress and stimulus responses (Supplementary Fig. 6a), consistent with the observed restricted plant growth under iron deficiency. A subset of these up-regulated genes was also categorized into the KEGG pathway of "phenylpropanoid biosynthesis" (Fig. 4b), from which the iron-mobilizing simple coumarins are derived, thus reflecting a typical adaptive mechanism to iron scarcity. Meanwhile, several down-regulated genes in iron-deficient Col-0 were enriched in GO terms and KEGG pathways related to flavonoid metabolism and biosynthesis (Fig. 4b, and Supplementary Fig. 6b). These findings collectively suggested a strategic reallocation in the biosynthesis of SMs under different external pressures. Since both flavonoids and simple coumarins are produced via the phenylpropanoid pathway[37], it seemed plausible that plants down-regulated flavonoid production to prioritize the synthesis of iron-scavenging coumarins.

WT-NyZ480 inoculation alone also triggered 575 DEGs in Col-0 roots (Fig. 4a), despite only limited bacterial colonization being observed (Fig. 3a, and Supplementary Fig. 5a). Among these, the 300 up-regulated genes were predominantly enriched in GO terms involved in response to external or biotic stimulus (Supplementary Fig. 6a), whereas the 275 down-regulated genes showed enrichment in metabolic and biosynthetic processes, and enzyme activities (Fig. 4b, and Supplementary Fig. 6b). These findings indicated that even limited colonization of WT-NyZ480 exerted non-negligible effects on Arabidopsis growth. This may account for the partial growth impairment observed in iron-sufficient plants following WT-NyZ480 inoculation (Fig. 3b).

Among all experimental groups, the combinatorial treatment of iron deficiency and WT-NyZ480 inoculation induced the most pronounced transcriptional reprogramming in Col-0 roots, with 844 DEGs detected (Fig. 4a). Among these, 697 genes were up-regulated, and 147 genes were down-regulated. Notably, GO enrichment analysis revealed that the up-regulated genes were categorized into GO terms associated with plant response to biotic stimulus, and defense response to bacterium (Supplementary Fig. 6a). In parallel, KEGG enrichment analysis indicated the activation of the "plant-pathogen interaction" pathway under iron-deficient conditions upon WT-NyZ480 inoculation (Fig. 4b). These observations may be attributed to the significantly enhanced colonization of Col-0 by WT-NyZ480, as excessive bacterial proliferation was likely to have activated plant defense machinery and, in turn, compromised plant fitness. In addition to intensified defense response, the biosynthetic flux of simple coumarins was seemingly amplified, evidenced by the significant enrichment of up-regulated genes in the "phenylpropanoid biosynthesis" pathway. This suggested that WT-NyZ480 colonization exacerbated iron stress in iron-deficient plants, thereby increasing the demand for iron-mobilizing simple coumarins. As for the down-regulated genes, results showed that KEGG pathways related to protein processing and GO terms in association with specialized metabolites biosynthesis and detoxification were enriched (Fig. 4b, and Supplementary Fig. 6b), indicating that normal physiological functions and metabolic activities in Col-0 were likely suppressed. This transcriptional suppression may explain the more severe growth impairment in iron-deficient Col-0 with WT-NyZ480 inoculation (Fig. 3b).

To sum up, WT-NyZ480 inoculation induced transcriptional changes in Col-0 despite its varying colonization efficiency under different iron conditions. Nevertheless, when inoculated under iron deficiency, the enhanced colonization of WT-NyZ480 likely exacerbated the iron stress of Col-0 and triggered more intense responses. Thus, transcriptomic evidence substantiated that the detrimental

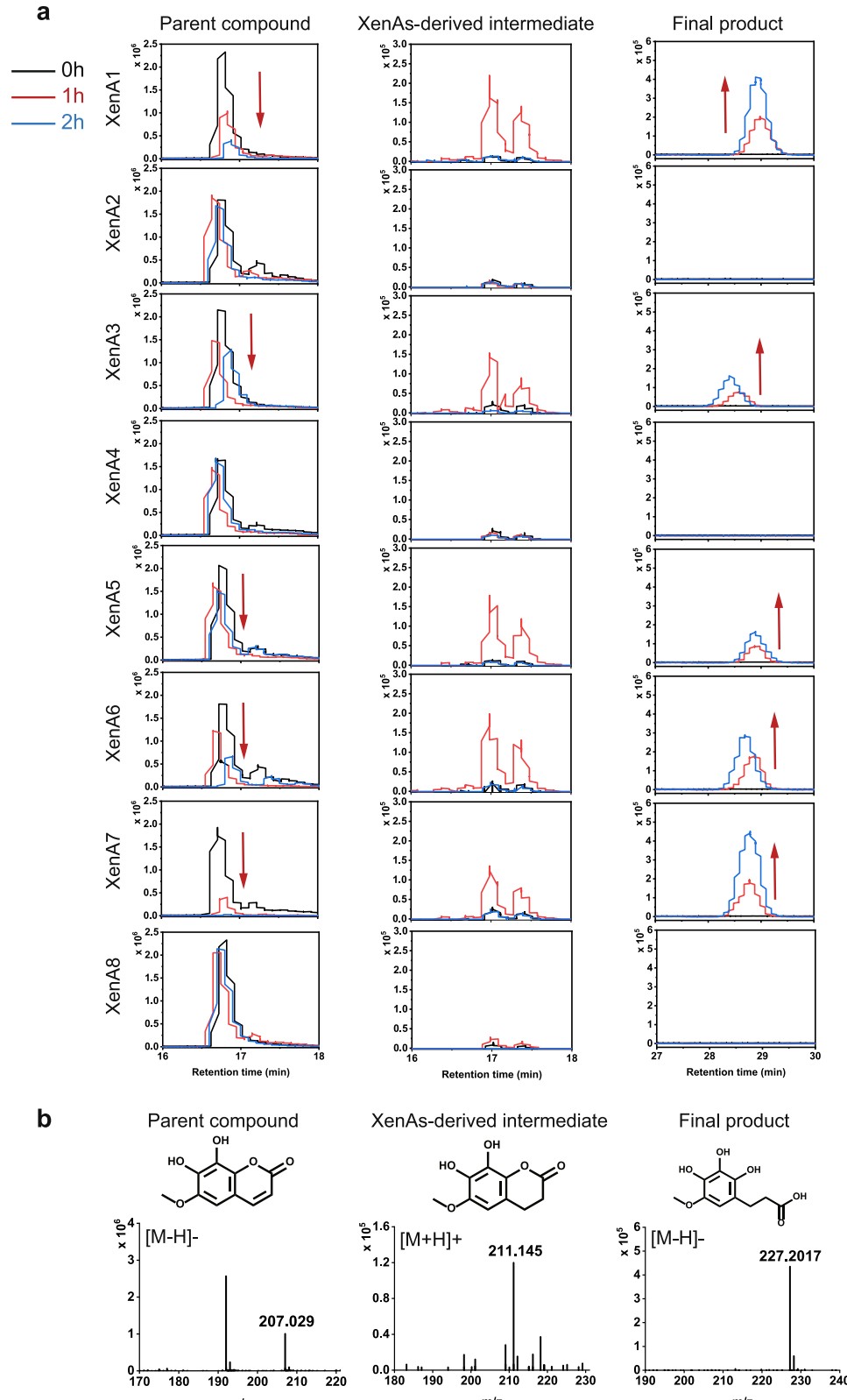

**Fig. 2 | NyZ480 harbors redundant XenA catalyzing simple coumarin degradation. a** Time course of FRA biotransformation by *E. coli* cells containing each of the eight heterologously expressed XenA enzymes (XenA1-8). Reactions were monitored over two hours by extracted ion chromatograms (EICs) for FRA, a transient intermediate, and the final hydrolysis product. Red arrows indicate the decrease of FRA and the corresponding increase of the final product. **b** Mass spectrometric characterization of all chemicals involved. Shown are the authentic FRA standard (negative ion mode), the proposed transient intermediate generated by XenA catalysis (positive ion mode), and the final product resulting from its spontaneous hydrolysis (negative ion mode).

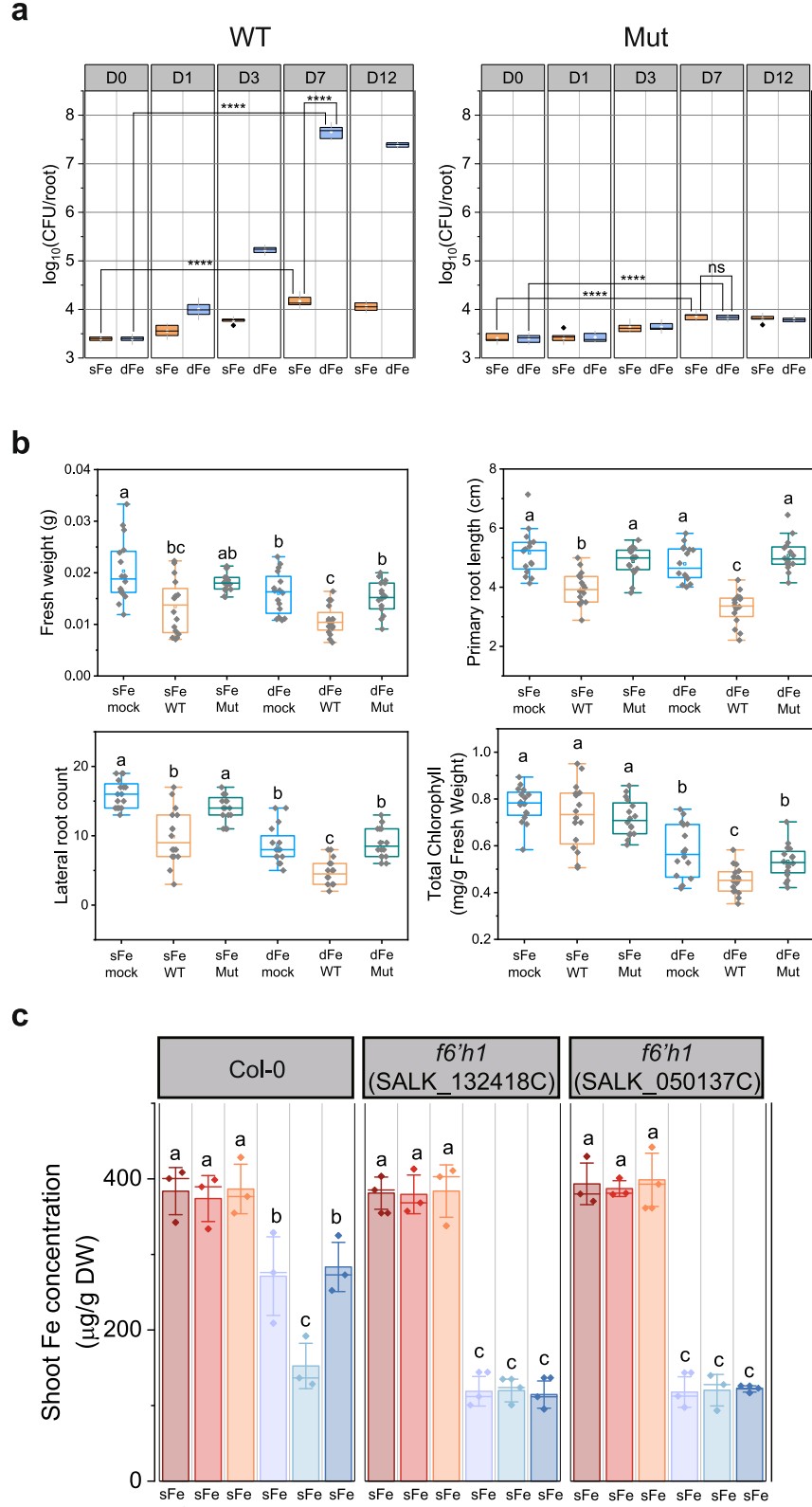

effects of WT-NyZ480 on plant fitness were specifically intensified under iron-deficient conditions.

### NyZ480 fails to establish enhanced colonization on Arabidopsis *f6'h1*

To investigate whether the enhanced colonization of Arabidopsis roots by WT-NyZ480 is exclusively dependent on the plant's secretion of simple coumarins at physiological levels, further experiments were conducted using Arabidopsis *f6'h1* (hereafter, *f6'h1*), a mutant defective in simple coumarin biosynthesis[11]. Due to its inability to produce iron-mobilizing simple coumarins, *f6'h1* exhibited severely impaired growth under iron-deficient conditions, accompanied by noticeable leaf chlorosis (Supplementary Fig. 7). Quantitative analysis using LC-MS also confirmed that, aside from limited amounts of coumarin

**Fig. 3 | NyZ480 significantly colonizes iron-deficient Arabidopsis Col-0 and compromises plant iron acquisition and overall fitness. a** Log$_{10}$-transformed value of Colony Forming Unit (CFU) per root. Plants were grown gnotobiotically for 14 days on iron-sufficient (sFe, 100 μM Fe$^{2+}$) or iron-deficient (dFe, 10 μM Fe$^{3+}$) ½ MS agar plates, and 20 μL of wild-type (WT) or mutant (Mut, namely NyZ480Δ8-*xenA*Δ*mhpB*) NyZ480 suspension (OD$_{600}$ = 0.003) was inoculated along the plant roots. Samples were taken at 0, 1, 3, 7, and 12 days post-inoculation (DPI) for CFU counting. Each treatment comprised three biological samples (*n* = 3), and three technical replicates were employed for each sample. Statistical differences between Day 0 and Day 7 within each treatment (sFe or dFe), and between sFe and dFe at Day 7 were determined by two-tailed Student's *t*-test (*p* < 0.05, ***p* < 0.001, *****p* < 0.0001; ns, not significant). **b** Phenotypic characterization of Col-0 following different treatments. Two-week-old plants grown under sFe or dFe conditions were inoculated with WT or Mut NyZ480 (OD$_{600}$ = 0.003), or with 10 mM MgCl$_2$ (mock).

Sampling was conducted at 7 DPI (*n* = 16, each treatment comprised 16 seedlings), and fresh weight, primary root length, lateral root count, and total chlorophyll were recorded. **c** Shoot iron concentration in Col-0 and *f6′h1* under different treatments. Two-week-old Arabidopsis plants were transferred to sFe or dFe ½ MS plates, and inoculated with 20 μL of WT or Mut NyZ480 suspension (OD$_{600}$ = 0.003). Control groups were treated with 10 mM MgCl$_2$ (mock). Shoot tissues were harvested at 7 DPI for quantification of iron concentration by ICP-MS. Each treatment consisted of three biological replicates (*n* = 3), with each replicate comprising a pool of three individual seedlings. Error bars indicate ± SD of biological replicates. For **a**, **b**, boxplots display the median, 25th/75th percentiles, and minima/maxima. Statistical differences in **b**, **c** were determined by one-way ANOVA followed by Tukey's post hoc tes (*p* < 0.05). All exact *p*-values are provided in the Source Data file.

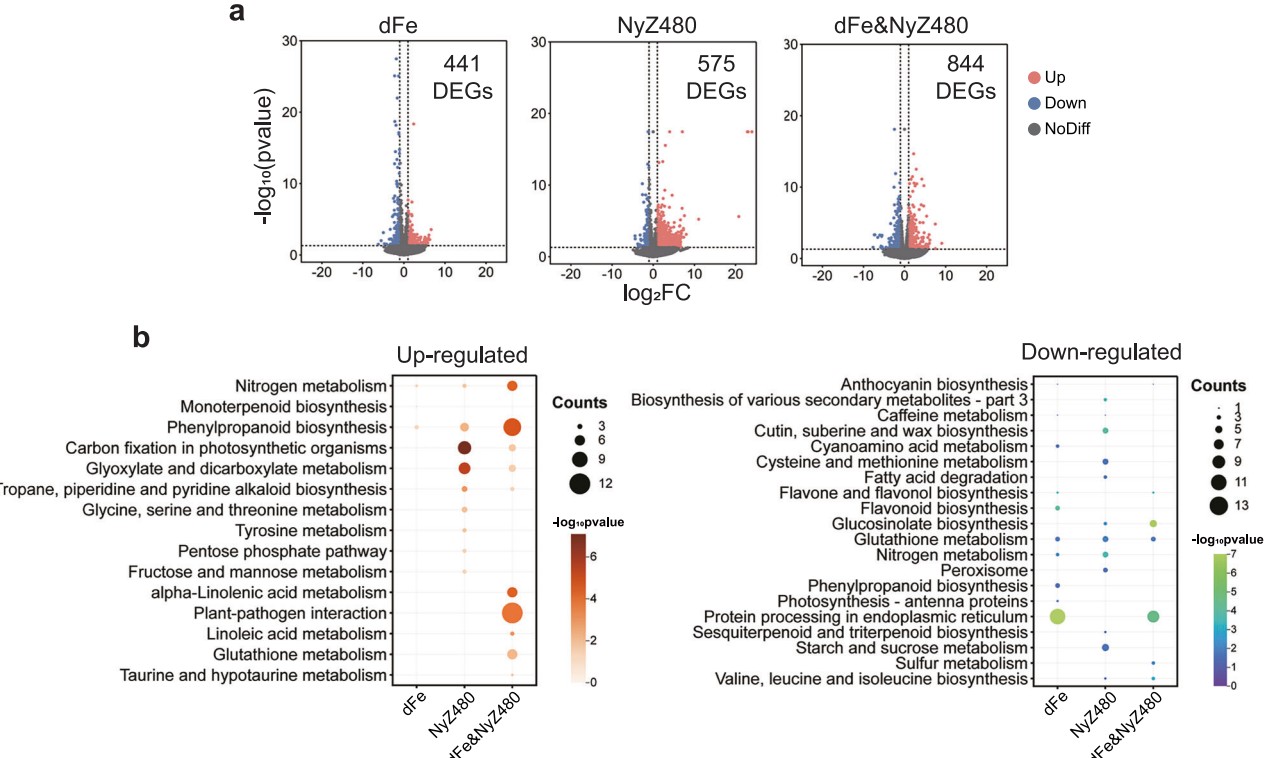

**Fig. 4 | Transcriptional changes of Arabidopsis Col-0 are caused by NyZ480 inoculation. a** Differentially expressed genes (DEGs) in Col-0 roots under different treatments. Transcriptomic profiles were compared among plants subjected to iron deficiency (dFe), NyZ480 inoculation (NyZ480), or a combinatorial treatment of iron deficiency and NyZ480 inoculation (dFe&NyZ480), with iron-sufficient and uninoculated plants as control. Two-week-old Arabidopsis Col-0 seedlings were uniformly picked from regular ½ MS agar plates, and subjected to the above-mentioned treatments (*n* = 3, each treatment included three biological replicates, one replicate comprised six individual seedlings). Sampling and RNA sequencing were performed seven days later. A two-tailed Wald test in DESeq2[55] was used to calculate *p*-values. Significantly up- or down-regulated genes (*p* < 0.05, |Log$_2$FC| > 1) are shown in red and blue dots, respectively. Numbers of DEGs relative to the control are indicated. **b** KEGG pathway enrichment analysis of DEGs from each treatment. Pathways significantly enriched among up- or down-regulated genes are shown. Dot size corresponds to the number of genes in each pathway. A one-tailed hypergeometric test in ClusterProfile[56] was used to calculate *p*-values, which were converted to -log$_{10}$(*p*-values) and represented by the color scale of the dots.

(detected at approximately 20 ng/mL), *f6′h1* no longer exuded esculetin, fraxetin, and scopoletin under iron deficiency (Supplementary Data 1). When WT-NyZ480 was inoculated onto roots of *f6′h1*, CFU assays demonstrated that this strain grew faintly on iron-deficient plant roots, with the log$_{10}$(CFU/root) value reaching 4.5 at 7 DPI (Fig. 5a), significantly lower than that on iron-deficient Col-0. This restricted colonization was further corroborated by CLSM imaging using *rfp*-tagged WT-NyZ480 (Supplementary Fig. 5b). Together with the earlier findings in Col-0 (Fig. 3a, and Supplementary Fig. 5a), these observations strongly supported a direct correlation between the enhanced WT-NyZ480 colonization and the production of simple coumarins by Arabidopsis under iron deficiency.

Examination of the impact of WT-NyZ480 on *f6′h1* under different iron conditions revealed that bacterial inoculation negatively affected plant fresh weight, primary root length, and lateral root count of iron-sufficient *f6′h1*, but did not alter total chlorophyll concentration (Fig. 5b) and shoot iron concentration (Fig. 3c). This was consistent with our observations in iron-sufficient Col-0, further demonstrating the general adverse effects of WT-NyZ480 on Arabidopsis fitness. Nevertheless, under iron deficiency, WT-NyZ480 no longer exerted discernible negative effects on *f6′h1* plants (Fig. 5b), which was notably different from that on Col-0 (Fig. 3b). This discrepancy may be explained by two interconnected factors. First, due to its impaired simple coumarin biosynthesis, *f6′h1* was found to develop more severe

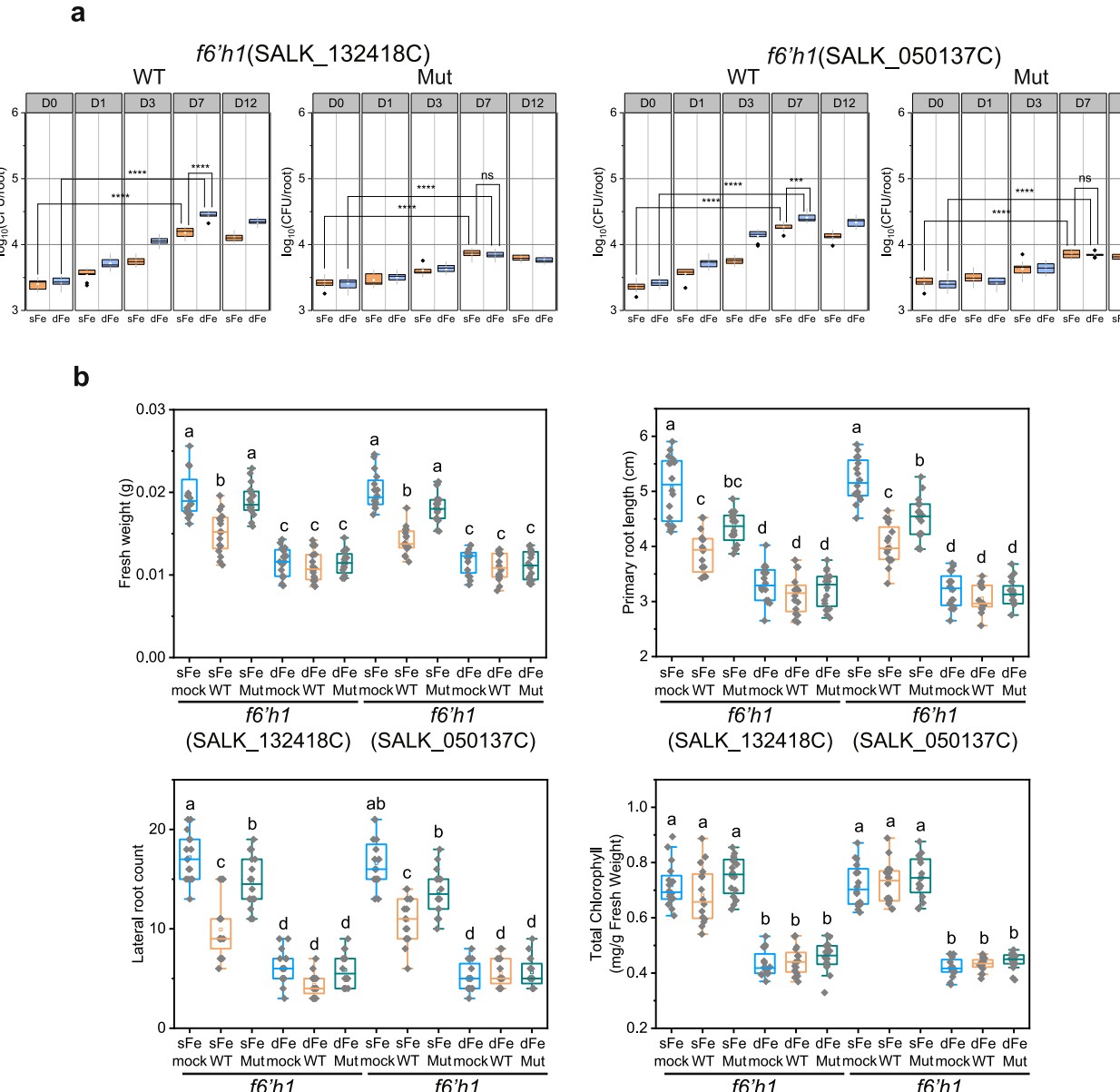

**Fig. 5 | WT- and Mut-NyZ480 fail to establish enhanced colonization on Arabidopsis mutant *f6'h1*. a** Log$_{10}$-transformed value of CFU per root. Arabidopsis *f6'h1* (SALK_132418C, SALK_050137C) plants were grown gnotobiotically for two weeks on sFe or dFe ½ MS agar plates, and 20 μL of WT- or Mut-NyZ480 suspension (OD$_{600}$ = 0.003) was inoculated along the plant roots. Samples were taken at 0, 1, 3, 7, and 12 DPI for CFU counting. Each treatment comprised three biological samples (*n* = 3), and three technical replicates were employed for each sample. Statistical differences between Day 0 and Day 7 within each treatment (sFe or dFe), and between sFe and dFe at Day 7 were determined by two-tailed Student's *t*-test

(*$p < 0.05$, ***$p < 0.001$, ****$p < 0.0001$; ns, not significant). **b** Phenotypic characterization of *f6'h1* following different treatments. Two-week-old plants grown under sFe or dFe conditions were inoculated with WT- or Mut- NyZ480 (OD$_{600}$ = 0.003), or with 10 mM MgCl$_2$ (mock). Sampling was conducted at 7 DPI (*n* = 16, each treatment comprised 16 seedlings), fresh weight, primary root length, lateral root count, and total chlorophyll were recorded. Statistical differences were determined by one-way ANOVA followed by Tukey's post hoc test (*p* < 0.05). All exact *p*-values are provided in the Source Data file. Boxplots in **a**, **b** display the median, 25th/75th percentiles, and minima/maxima.

growth defects under iron deficiency than Col-0. Second, due to the inability to produce simple coumarins, WT-NyZ480 failed to establish enhanced colonization on iron-deficient *f6'h1* roots. Consequently, the limited bacterial presence on *f6'h1* failed to further impair plant fitness to the same extent as on Col-0.

### Coumarin catabolic genes are essential for NyZ480 colonization on Arabidopsis

Having demonstrated, using the Arabidopsis mutant *f6'h1*, that host production of simple coumarins was indispensable for the enhanced colonization of WT-NyZ480 under iron deficiency, we employed

NyZ480 mutants to further investigate whether the bacterial utilization and detoxification capacity against these phytocompounds underpinned its proliferation on iron-stressed plant roots.

Serving as the primary degradation-initiating genes, all the redundant *xenA*s were sequentially inactivated in NyZ480 to construct mutant strains (Supplementary Fig. 8). Bacterial growth of the generated mutants was assessed with coumarin or fraxetin as the sole carbon source. Results indicated that disruption of all five functional *xenA* genes (*xenA1, xenA3, xenA5, xenA6,* and *xenA7*) did not impair bacterial growth on coumarin (Supplementary Fig. 9a), suggesting the existence of uncharacterized genes responsible for coumarin transformation in

NyZ480. To obtain a growth-abolished mutant, the previously validated gene *mhpB*[34], essential for coumarin downstream catabolism, was additionally knocked out. The resultant strain, NyZ480Δ5*xenAs*Δ*mhpB*, eventually failed to grow on coumarin. In contrast, when fraxetin was supplied as the sole carbon source, bacterial growth was progressively impaired with the sequential knockout of *xenA* genes (Supplementary Fig. 9b). The NyZ480Δ5*xenAs*Δ*mhpB* mutant also completely lost its capacity to utilize fraxetin. This indicated that the redundancy of *xenA* genes seemed to be integral to fraxetin-supported growth, and the catabolism of simple coumarins converged through a common downstream pathway via the *mhp* cluster, aligning with the NyZ480 transcriptomic data (Fig. 1e).

Although esculetin and scopoletin are not utilized as the sole carbon sources by NyZ480 (Fig. 1d), our data indicated that multiple XenAs have catalytic activity against these two compounds (Fig. 2, and Supplementary Fig. 4). It was hypothesized that the robust transformation of esculetin and scopoletin by redundant *xenAs* served as a defensive detoxification mechanism in NyZ480, given the reported antimicrobial properties of these coumarins[16,30]. To test this, detoxification capacity of NyZ480 mutants was assessed on TSA plates supplemented with different simple coumarins. Results indicated that the growth of mutant strains on TSA plates containing esculetin or scopoletin was progressively attenuated following the successive deletions of *xenA* genes (Supplementary Fig. 9c), suggesting that the redundant *xenA* genes in NyZ480 enabled bacterial growth in the presence of antimicrobial simple coumarins by mediating their detoxification. Collectively, *xenA* redundancy equips NyZ480 with the dual capacity to utilize certain simple coumarins as carbon sources and to detoxify the antimicrobial activity of others.

When the constructed NyZ480 mutant strains were all cultured in LB medium, their growth showed no difference compared to the wild type (Supplementary Fig. 10). This indicated that the knockout of catabolic genes specifically disrupts the strain's ability to degrade simple coumarins without affecting its general growth on standard carbon sources. Subsequently, we selected NyZ480Δ8*xenAs*Δ*mhpB* (hereafter referred to as Mut-NyZ480, with all eight *xenA* genes and the *mhpB* gene knocked out) as a representative NyZ480 mutant, tagged with *rfp*, and inoculated on the Arabidopsis roots. Under both iron-sufficient and iron-deficient conditions, Mut-NyZ480 exhibited lower colonization than WT-NyZ480 on both Col-0 and *f6′h1* Arabidopsis, as demonstrated by CFU counting (Figs. 3a and 5a, the $\log_{10}$(CFU/root) values all remained below 4 at 7 DPI) and CLSM analysis (Supplementary Fig. 5). As previously noted, markedly enhanced colonization of WT-NyZ480 was observed exclusively on iron-deficient Col-0, whereas limited bacterial growth still occurred on both Col-0 and *f6′h1*, regardless of their iron status. This pattern stood in stark contrast to that observed for Mut-NyZ480, indicating that the coumarin catabolic genes in NyZ480 are essential not only for the opportunistic, enhanced colonization on iron-deficient Col-0 but also for the limited growth of NyZ480 on Arabidopsis, irrespective of plant genotype or iron status. It was therefore reasonable to speculate that enzymes encoded by these catabolic genes may also act on other non-specific SMs secreted by Arabidopsis roots, thereby enabling this limited bacterial colonization.

Because the colonization capacity of Mut-NyZ480 was compromised, the growth defects observed in Col-0 and *f6′h1* after WT-NyZ480 treatment were partially alleviated when Mut-NyZ480 was inoculated instead (Figs. 3b and 5b). In particular, unlike WT-NyZ480, inoculation with Mut-NyZ480 did not reduce iron levels in iron-deficient Col-0 (Fig. 3c), demonstrating that the disrupted plant iron acquisition was likely caused by WT-NyZ480's degradation of simple coumarins.

Collectively, these results demonstrated that the genetic determinants of coumarin catabolism in NyZ480 promoted its growth on Arabidopsis roots, particularly the pronounced colonization on iron-

deficient Col-0. Furthermore, the growth impairment of Arabidopsis was linked to the extent of bacterial colonization, which was most pronounced in iron-deficient Col-0 following WT-NyZ480 inoculation.

## Degraders of simple coumarins are prevalent in the environment

Considering that plant vitality, which relies on simple coumarins for iron acquisition under deficiency, could be compromised by coumarin-degrading rhizobacteria, simple coumarin catabolic genes were surveyed among environmental bacteria. Among 51,445 annotated bacterial genomes retrieved from the IMG/M database[38], up to 84% (43,013) of them contain *xenA*, spanning 34 phyla (Fig. 6a, and Supplementary Data 2), with the exception of certain phyla from unusual habitats such as abyssal sediments and deep-sea hydrothermal vents (e.g., *Calditrichota*, *Candidatus*, *Absconditabacteria*, *Dictyoglomota*, *Kiritimatiellota*, and *Thermomicrobiota*). These results indicated that the degradation-initiating *xenA* gene is present in bacteria across a wide range of ecosystems. The *xenA* gene is particularly prevalent in bacteria from aerial, terrestrial, and plant-associated environments (over 96%), and its occurrence in bacteria from other sources remains above 73% (Fig. 6b, and Supplementary Data 3), underscoring its universal distribution. In addition, gene redundancy of *xenA* is prevalent, with 79% of bacterial genomes harboring more than one copy of *xenA*, and 25% having five or more homologs (Supplementary Data 2). Bacteria harboring 10 or more copies of *xenA* predominantly belong to *Pseudomonadota*[39], one of the most abundant soil-dwelling phyla, and over half of them are from the genus *Pseudomonas* (Supplementary Data 2). Notably, the plant-associated *Rhodococcus* sp. LB1 was found to carry the most *xenA* homologs, up to 25 gene copies. From these results, it was proposed that the high prevalence and redundancy of the degradation-initiating *xenA* gene in soil-associated bacteria may reflect an evolutionary adaptation to the sustained exposure of simple coumarins, given their widespread synthesis and secretion across the plant kingdom.

The presence of *xenA* in bacteria suggests a capability for the initial transformation of simple coumarins. This breakdown of the coumarin structure is likely sufficient to detoxify these antimicrobial plant-derived SMs, thereby conferring bacterial resistance. However, a complete set of catabolic genes is required for bacterial proliferation utilizing simple coumarins, subsequently disrupting iron acquisition and compromising plant fitness. Hence, the presence of the remaining coumarin catabolic genes was also investigated in environmental bacteria. Compared to *xenA*, the genes *couC* and *mhpBCDEFRT* cluster were rarely found in bacteria (Fig. 6b). In particular, *couC*, which is essential for coumarin catabolism in NyZ480[34], showed the lowest average occurrence frequency across bacteria from various sources. It was thus hypothesized that bacteria capable of utilizing simple coumarins for growth are likely uncommon in the environment. This was confirmed by subsequent analyzes, which identified only 23 bacterial strains harboring the complete genetic determinants required for coumarin catabolism (Fig. 6c). Those potential simple coumarin utilizers all belong to the phylum *Pseudomonadota*, including the genera *Paraburkholderia*, *Pandoraea*, *Ralstonia*, *Pseudomonas*, *Burkholderia*, and *Cupriavidus*, with most originating from soil environments. Similar to NyZ480, each of the 23 strains carries five or more copies of *xenA* (Supplementary Data 4). Moreover, the *couC* gene is not clustered with other catabolic genes (Supplementary Data 5), whereas the *mhpBCDEFRT* genes, responsible for downstream catabolism of coumarin, generally form a gene cluster in these bacteria, except for five strains with scattered *mhp* genes (Fig. 6c, and Supplementary Fig. 11). The genetic organization of the *mhp* cluster is primarily conserved among phylogenetically related strains, suggesting that the acquisition of these genes is likely achieved via vertical transmission rather than horizontal gene transfer.

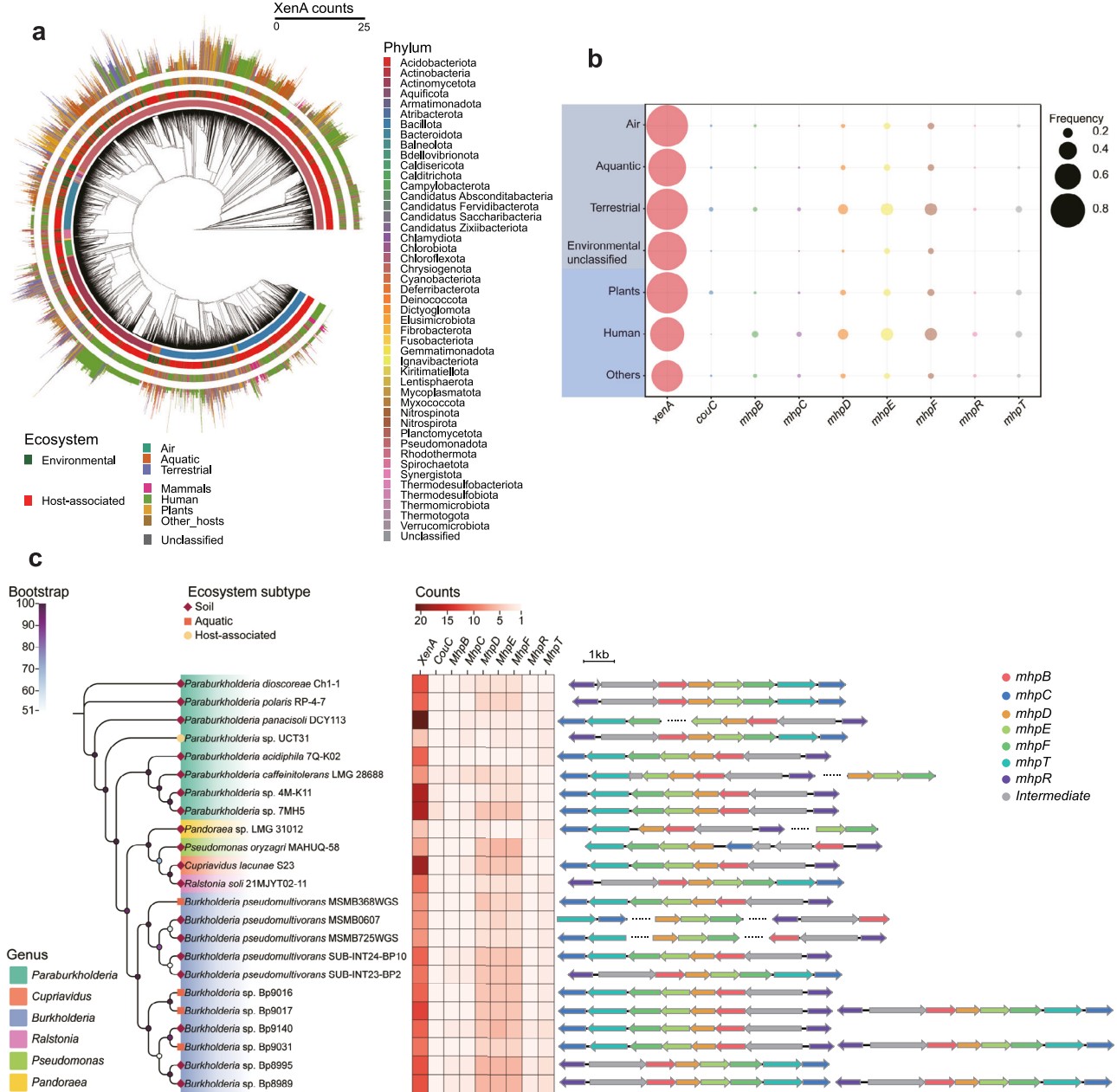

**Fig. 6 | Distribution of simple coumarin catabolic genes in environmental bacteria. a** A phylogenetic tree of 51,445 bacterial genomes downloaded from the IMG/M database. The concentric rings (from inner to outer) indicate bacterial phylum, broad ecosystem classification, specific ecosystem category, and the *xenA* homolog copy number, respectively. **b** Occurrence frequency of simple coumarin catabolic genes in bacterial genomes from different sources is represented by dot size. Categories of bacterial origin on a light purple background are environmental-associated; those on a light blue background indicate host-associated origins. **c** Phylogenetic relationship, gene copy number, and genomic organization of simple coumarin catabolic gene clusters in 23 bacterial isolates harboring the complete simple coumarin catabolic pathway. Phylogenetic analysis was performed with GTDB-Tk (v2.4.0)[65], based on a concatenated alignment of 120 bacterial marker genes.

## Discussion

In this study, we demonstrate that the rhizobacterium *Pseudomonas* sp. strain NyZ480 possesses robust catabolic and detoxification capabilities for simple coumarins, a key class of plant SMs synthesized and secreted to facilitate iron acquisition and pathogen defense. This capacity enables NyZ480 to establish enhanced colonization on the iron-deficient, coumarin-secreting Arabidopsis, thereby severely impairing plant iron acquisition and overall fitness. Although the degradation-initiating gene *xenA* is prevalent and often redundant across many environmental bacteria, the complete set of simple coumarin catabolic genes is rarely co-present in individual bacterial

genomes. We propose that the phenomenon of bacterial degradation of plant-derived simple coumarins leading to impaired plant growth should be emphasized in natural settings.

Numerous studies have highlighted the essential role of simple coumarins in facilitating plant iron uptake under iron-limited conditions[8,10,36,40], as well as their antimicrobial activity against pathogens[9,18,30]. Previous quantitative analyzes have shown that iron deficiency induces synthesis and exudation of simple coumarins in Arabidopsis, including esculetin[11], fraxetin[12], and scopoletin[14], with scopoletin being the most abundantly secreted in hydroponic culture[14]. Our findings are largely consistent with these reports, except

that no increased accumulation of esculetin was observed. Catechol coumarins, including esculetin, fraxetin, and sideretin[7], have been verified to mobilize iron over a range of pH conditions[10], as their redox-active feature enables the reduction of $Fe^{3+}$ to the more bioavailable $Fe^{2+}$. Notably, however, these catechol coumarins are exuded only sparingly by iron-deficient Col-0 compared with scopoletin, which lacks the catechol moiety. Given that scopoletin has been reported to profoundly shape the rhizospheric microbiota[14], we hypothesize that simple coumarins mediate plant iron acquisition not only through direct reduction or chelation of iron, but also by selectively modulating rhizospheric microbial communities, thus potentially enriching beneficial microbes that promote iron uptake while inhibiting iron competitors. Surprisingly, our study demonstrates that the rhizosphere-isolated NyZ480 degrades all the above-mentioned simple coumarins, suggesting that the selection imposed by plant SMs on microbes can be overcome by the microbes themselves, thereby leading to outcomes that are detrimental to plant growth.

From the Arabidopsis transcriptome, the combinatorial treatment of iron deficiency and NyZ480 inoculation further enhances the expression of several key simple coumarin biosynthesis genes, including *4cl*, *f6′h1*, *cosy*, *s8h* and *cyp82c4*, compared with iron deficiency alone (Supplementary Fig. 12a, c). This indicates that NyZ480 inoculation intensifies the plant's transcriptional responses to iron deficiency, promoting increased synthesis and exudation of simple coumarins. However, following NyZ480 inoculation, simple coumarins are no longer detectable in the hydroponic medium, consistent with their degradation by this strain. Since NyZ480 not only degrades but also grows on simple coumarins, this continual degradation and utilization massively depletes the iron-mobilizing coumarins, likely further compromising plant uptake of the limited iron sources. This, in turn, triggers an amplified biosynthesis and secretion flux of simple coumarins. These newly exuded compounds, nevertheless, are again degraded and utilized by NyZ480. Hence, the persistent iron scarcity of plants continuously fuels NyZ480 growth, creating a "vicious circle" that traps the plants in an impaired state. This proposed framework explains why simple coumarins biosynthesis genes are more significantly up-regulated by NyZ480 inoculation under iron deficiency, despite the absence of detectable accumulation of these compounds in the medium. Together, these findings illustrate that plant-associated bacteria can exploit plant-derived SMs under certain conditions to enhance their own growth at the expense of host fitness.

Genetic redundancy, a common feature in prokaryotes[41,42], gives microorganisms an advantage in mitigating the metabolic burdens imposed by structural analogs in the environment[43]. Here, the degradation of coumarin[33], esculetin, fraxetin, and scopoletin is catalyzed by enzymes encoded by multiple *xenA* homologs in NyZ480. While some *xenA* genes (e.g., *xenA5*, *xenA6*) are constitutively expressed, others (e.g., *xenA1*, *xenA3*, *xenA7*) are induced by simple coumarins. This indicates that NyZ480 has evolved sophisticated catabolic redundancy to accommodate these SMs in plant-associated environments. Consequently, this strain exhibits versatile growth on multiple simple coumarins as sole carbon sources, and also efficiently detoxifies these antimicrobial phytocompounds. These findings illustrate a genetically efficient strategy employed by bacteria to thrive in the rhizosphere, where structurally related SMs are enriched. In contrast to the redundant and scattered degradation-initiating *xenA* genes, the downstream genetic determinants for simple coumarin degradation converge on the single *mhp* cluster. Since this gene cluster was initially characterized for its role in catabolizing 3-(3-hydroxyphenyl) propionate[44], an aromatic compound derived from lignin, we therefore propose that microbial degradation of diverse natural phytocompounds may adopt assemblies of distinct upstream catabolic routes and convergent downstream pathways. This modular organization likely enables microorganisms to maximize substrate utilization and gain competitive growth advantages in plant-associated niches.

Furthermore, the compromised colonization of Mut-NyZ480 on Arabidopsis implies that both the redundant *xenA* genes and the *mhp* cluster are crucial for NyZ480's utilization of other non-specific plant-derived metabolites, particularly the latter. Given the previously established role of the *mhp* cluster in degrading lignin-derived compounds and its involvement in simple coumarin catabolism here, we propose that this catalytically promiscuous cluster may mediate the catabolism of diverse plant-derived molecules. Accordingly, WT-NyZ480 mildly colonizes Arabidopsis even in the absence of secreted simple coumarins (e.g., Col-0 of sFe; *f6′h1* of sFe and dFe), thereby causing host transcriptional reprogramming and compromising plant fitness.

To understand the mechanism underlying the bacterial detriment to plants, subsequent genome mining revealed the presence of a type III effector (*orf736*) in NyZ480. Such effectors typically subvert host immunity to facilitate bacterial invasion and colonization[45]. However, this orphaned type III effector is also found to be a shared genomic feature of several commensal *Pseudomonas* strains[46] associated with plants. Hence, the molecular mechanism underlying the detrimental effects of NyZ480 on Arabidopsis remains elusive. What is unequivocally demonstrated here is that the compromised plant growth is closely linked to bacterial colonization. In particular, the iron deficiency-induced enhanced colonization of NyZ480 further perturbs plant iron acquisition through continuous degradation of simple coumarins, ultimately resulting in more severe growth damage to iron-deficient plants.

Interactions between rhizobacteria and their host plants are ubiquitous in the rhizosphere, where plant-derived SMs play pivotal roles. Bacteria harboring specific catabolic genes for particular SMs are likely recruited or enriched by plants. For instance, the *ifc* gene cluster, which enables the utilization of plant-derived isoflavones, is frequently detected in bacterial strains isolated from *legumes*[47]. Similarly, the complete inositol-degrading *iol* locus is conserved among superior root colonizers of Arabidopsis[48]. According to our bioinformatic surveys, simple coumarin utilizers are rare among environmental bacteria due to the infrequent co-occurrence of the complete genetic determinants of coumarin catabolism in bacterial genomes, likely indicating a relatively low incidence of impaired plant growth caused by microbial over-proliferation using these plant-derived SMs. Nevertheless, in addition to NyZ480, another bacterial strain exhibiting a phenotype identical to that of NyZ480 was later isolated from the Arabidopsis rhizosphere. This strain, NyZ490, also belongs to the genus *Pseudomonas* (its 16S rRNA gene identity with NyZ480 is 99.07%) and was similarly enriched and screened using coumarin as the sole carbon source (Supplementary Fig. 13a). NyZ490 demonstrates the same capacity to degrade simple coumarins (Supplementary Fig. 13b) and significantly colonizes iron-deficient Col-0 roots (Supplementary Fig. 13c), thereby impairing iron absorption and exacerbating growth defects in iron-deficient plants (Supplementary Fig. 13d, e). Thus, the simple coumarin biodegradation–associated plant–microbe interaction revealed in this study is not restricted to a single isolate and may represent a conserved trait within certain *Pseudomonas* species. Given the cosmopolitan distribution and ecological significance of *Pseudomonas*, coumarin-degrading strains within this genus that impair plant iron acquisition may represent a prevalent phenomenon in natural soil communities.

Although the overall abundance of simple coumarin-utilizing bacteria remains to be fully established, it is evident that the degradation-initiating gene *xenA* is both prevalent and highly redundant among environmental bacteria. This widespread presence suggests that detoxification of simple coumarins may be a frequent strategy, thereby enabling microorganisms to detoxify these antimicrobial SMs and thrive by using other carbon sources. This dynamic observed here resembles a plant-microbe "arms race", similar to a recently reported case in maize, where root-derived benzoxazinoids

are metabolized by diverse and abundant rhizobacteria[49]. Even though simple coumarins can be massively detoxified by XenA, severe detriment is more likely caused to plants when these phytocompounds are completely utilized by rhizobacteria for enhanced colonization. Here, we propose that the sustained degradation of simple coumarins by NyZ480 perpetuates the elimination of these iron-mobilizing SMs from the rhizosphere. This persistent removal may create a feedback loop in which the plant secretes even more simple coumarins to compensate; these additional coumarins are, in turn, degraded again by NyZ480, thereby progressively compromising plant iron acquisition and overall health.

Previous studies have often highlighted beneficial outcomes following bacterial catabolism of plant SMs. For instance, the flavonoids attracted *Aeromonas* sp. H1 enhances plant dehydration resistance[50], and the purine-enriched *Pseudomonas* improves wild soybean growth under salt stress[51]. Here, we demonstrate a contrasting, detrimental scenario in which a rhizobacterium opportunistically enhances its root colonization by degrading and utilizing simple coumarins secreted under iron stress, ultimately impairing host plant fitness. These findings underscore that specific microbes may subvert plant's adaptive mechanism into an exploitable vulnerability. Given that iron deficiency challenges global agricultural productivity, greater attention should be directed to the potential disruption of plant iron acquisition by such coumarin-degrading rhizobacteria. We acknowledge that interactions in native soil microbiomes are likely more intricate due to competition and cooperation among diverse microbes. The generalizability of this monoculture-based mechanism requires further validation. Nonetheless, this study provides a crucial proof-of-concept that such a microbial exploitative strategy can occur.

## Methods

### Plant species and growth conditions
Arabidopsis Col-0 and *f6'h1* (SALK_132418C, SALK_050137C) were used as the plant materials in this study. For sterilization, seeds were immersed in 70% ethanol for 15 min, followed by 15 min in 50% bleach solution, and washed three times with sterile distilled water. Sterilized seeds were stratified at 4 °C in the dark for 2 days, and subsequently sown on ½ Murashige and Skoog (½ MS, pH 5.8) agar plates supplemented with 0.5% sucrose (wt/vol) (ingredients of MS except iron: 20.61 mM $NH_4NO_3$, 100.27 μM $H_3BO_3$, 2.99 mM $CaCl_2$, 0.11 μM $CoCl_2 \cdot 6H_2O$, 0.10 μM $CuSO_4 \cdot 5H_2O$, 100 μM $Na_2EDTA \cdot 2H_2O$, 1.50 mM $MgSO_4$, 100.00 μM $MnSO_4 \cdot H_2O$, 1.03 μM $H_2MoNaO_5 \cdot 2H_2O$, 5.00 μM KI, 18.79 mM $KNO_3$, 1.25 mM $KH_2PO_4$, 29.91 μM $ZnSO_4 \cdot 7H_2O$, 26.64 μM glycine, 554.94 μM myo-inositol, 4.06 μM nicotinic acid, 2.43 μM pyridoxine·HCl, 0.30 μM thiamine·HCl). Plant culture conditions of iron deficiency and iron sufficiency were defined as supplementation with 10 μM $FeCl_3$ and 100 μM $FeSO_4$ to the ½ MS medium, respectively. Iron supplements were adjusted according to different assays conducted below. The plates were vertically positioned, and seedlings were grown at 22 °C under a 16 h light/8 h dark cycle, with photon flux of 100–150 μE m$^{-2}$ s$^{-1}$.

### Bacterial growth and biodegradation assays
Single colonies of wild-type NyZ480 and its mutants were inoculated in 5 mL Lysogeny broth (LB) liquid medium. After overnight culture at 30 °C, 180 rpm, bacterial cells were harvested (5000 × g, 5 min), and washed three times with Phosphate-Buffered Saline (PBS, pH 7.4). Then, the pelleted cells were resuspended in 5 mL minimal medium (MM, ingredients: 40 mM $Na_2HPO_4$, 22 mM $KH_2PO_4$, 7.5 mM $(NH_4)_2SO_4$, 0.01 mM $CaCl_2$, 0.2 μM $MnSO_4 \cdot H_2O$, 0.3 μM $CuSO_4$, 0.3 μM $ZnSO_4$, 1.6 μM $MgSO_4 \cdot 7H_2O$, 1.1 μM $FeSO_4 \cdot 7H_2O$) and used as seed culture.

Bacterial growth with simple coumarins as the sole carbons were monitored with the Bioscreen C system (Oy Growth Curves Ab Ltd, Helsinki, Finland). Culture conditions were set at 30 °C, with medium shaking intensity, and $OD_{600}$ values were recorded every 30 min. MM was used as a culture medium, and the culture mixture also included 1% inoculum of NyZ480 seed solution (vol/vol), 0.3 mM of the respective substrate. Every treatment was triplicated.

Biodegradation assays were conducted by adding 0.5 mM of coumarin, esculetin, fraxetin and scopoletin into each of 5 mL seed solution of NyZ480, and the mixtures were then incubated on a rotary shaker (30 °C, 180 rpm). Samplings were performed at 0, 1, 2, 4, 6 h. Supernatants of the sampled solutions were retained after centrifugation (16,000 × g, 5 min) and further filtered with 0.22-μm nylon membrane. The processed samples were stored at −20 °C for liquid chromatography-mass spectrometry (LC-MS, Agilent 1290 Infinity II / 6545 QTOF) analysis.

### Quantification of simple coumarins secreted by Arabidopsis
Two-week-old Arabidopsis seedlings that were grown on ½ MS agar plates containing different iron supplies were uniformly picked and transferred to six-well plates filled with 3 mL liquid ½ MS medium that was supplemented with iron as described above. Each well contained four seedlings, and three replicates were used for each treatment. After four more days of growth, the used culture media were harvested. After lyophilization, 300 μL of MeOH/H$_2$O (80:20, v/v) solvent was used to resuspend the freeze-dried media, followed by vortex and sonication for 5 min. The collected samples were filtered through 0.22-μm nylon membrane and stored at −20 °C for LC-MS (Agilent 1260 / 6470 QQQ) analysis.

### Heterologous expression of redundant *xenA* and whole cell biotransformation
DNA fragments of eight *xenA* homologs were amplified with specific primers (Supplementary Table 1), and then inserted into linearized pET-28a(+) to generate expression constructs (Supplementary Table 2), each of which was finally introduced into host bacterium *Escherichia coli* BL21(DE3). *E. coli* BL21(DE3) strains harboring different constructs were cultured in 50 mL LB medium (37 °C, 200 rpm) until the bacterial $OD_{600}$ value reached 0.6. Subsequently, 0.3 mM isopropyl-β-D-thiogalactopyranoside (IPTG) was added to trigger the expression of proteins, and was followed by 14 h incubation at 16 °C, 180 rpm. Then, bacterial cells were harvested (5000 × g, 5 min), thoroughly washed with PBS (pH 7.4) and resuspended in 5 mL MM ($OD_{600}$ values were around 15). Whole-cell biotransformation was initiated by adding 0.3 mM esculetin, fraxetin or scopoletin to the bacterial suspensions. Samples were taken every hour, and were rigorously mixed with an equal volume of acetonitrile. After centrifugation (16,000 × g, 5 min), supernatants were preserved and then filtered through 0.22-μm nylon membrane. The obtained samples were stored at −20 °C for LC-MS (Agilent 1290 Infinity II/6545 QTOF) analysis.

### Generation of NyZ480 mutants
For markerless gene knockout of *xenA1*, *xenA2*, *xenA3*, *xenA4*, *xenA5*, *xenA7*, *xenA8* and *mhpB* in NyZ480, the upstream and downstream DNA fragments of each target gene were amplified (Supplementary Table 3) and fused, respectively. Then, the generated fragments were integrated into the suicide plasmid pK18mobsacB to construct gene deletion vectors (Supplementary Table 2), which were subsequently introduced into the conjugative donor strain *E. coli* S17-1. Successful transformants of *E. coli* S17-1 were individually screened and cultured in 5 mL LB medium with kanamycin (50 μg/mL) overnight (37 °C, 180 rpm). Afterwards, bacterial cells were harvested, washed, and resuspended in 500 μL PBS (pH 7.4). Cultivation of NyZ480 was conducted as above-mentioned, and bacterial cells were also concentrated with PBS (pH 7.4). Isometric bacterial suspensions of *E. coli* S17-1 and NyZ480 (20 μL each) were mixed and pipetted onto the sterile filter paper (1.5 cm × 1.5 cm) placed on LB agar plates, which were incubated overnight at 30 °C for homologous recombination. Bacterial cells were

then washed down from the filter paper, diluted with PBS (pH 7.4) and spread on LB agar plates containing kanamycin (50 µg/mL) and ampicillin (100 µg/mL). For the verification of the first-round homologous recombination event, single colonies were picked and subjected to PCR with specific primers (Supplementary Table 4). The successful recombinants were again cultured overnight (30 °C, 180 rpm) in 5 mL liquid LB medium supplemented with kanamycin (50 µg/mL) and ampicillin (100 µg/mL). Bacterial cells were harvested (5000 × $g$, 5 min), thoroughly washed, and diluted with PBS (pH 7.4), which were then spread on LB agar plate containing 10% sucrose (wt/vol) to select the second-round homologous recombination. Finally, the grown colonies were screened for markerless knockout of the target genes using PCR and DNA sequencing (Supplementary Table 4). After markerless knockouts of all target genes were successfully achieved except for *xenA6*, insertional inactivation was conducted. A 500-bp DNA region within the *xenA6* gene was amplified and cloned into pK18mobsacB, and the constructed vector was then introduced into *E. coli* S17-1. The first-round homologous recombination was performed as described, and the *xenA6*-inactivated mutant was screened and verified as above (Supplementary Table 4). For multigene deletion in NyZ480, insertional inactivation of *xenA6* was always performed last.

## Monitoring NyZ480 colonization on Arabidopsis roots

The gene encoding red fluorescent protein (*rfp*) was inserted into linearized pBBR1MCS-2 or pRK415 (Supplementary Table 2) using a One Step cloning kit (Vazyme, Nanjing, China). The resulting plasmids, pMCS-rfp and pRK-rfp, were introduced into wild-type NyZ480 (WT) and its mutant NyZ480Δ8*xenAs*Δ*mhpB* (Mut) via electro-transformation, respectively. Positive transformants were selected on LB plates supplemented with kanamycin (50 µg/mL) for WT (pMCS-rfp) or tetracycline (100 µg/mL) for Mut (pRK-rfp). Bacterial suspensions of *rfp*-tagged WT and Mut-NyZ480 were prepared as aforementioned, and adjusted to OD$_{600}$ = 0.003 with 10 mM MgCl$_2$. Two-week-old Arabidopsis seedlings grown gnotobiotically on ½ MS plates were inoculated onto the roots with 20 µL of bacterial suspension; control plants received 20 µL of 10 mM MgCl$_2$. After inoculation, Arabidopsis continued to grow under the same culture conditions. Root samples were taken at 7 DPI, and washed with sterile distilled water. Then, imaging using confocal laser scanning microscopy (CLSM, Nikon Ni-E A1 HD25) was conducted to visualize *rfp*-tagged bacteria.

For CFU quantification, seedlings were collected at 0, 1, 3, 7, 12 DPI and transferred into 1.5 mL tubes with 1 mL PBS (pH 7.4), and vortexed vigorously for 2 min. Serial dilutions were plated (100 µL) onto LB plates, and CFUs were counted; only plates with 30–300 colonies were used for analysis.

## RNA sequencing experiment and data analysis

Transcriptomic analysis of NyZ480 under the induction of simple coumarins was conducted as follows: NyZ480 was cultured overnight in 500 mL MM supplemented with 2 mM glucose (30 °C, 180 rpm). Bacterial cells were pelleted (5000 × $g$, 5 min), thoroughly washed, and resuspended in 100 mL MM, which were then divided into two equal portions. Simple coumarins (a mixture of 0.1 mM each of esculetin, fraxetin and scopoletin) were added to one portion of the NyZ480 suspension as an induction group, and 0.5 mM glucose was added in the other portion as a control group. After an additional 8 h cultivation (30 °C, 180 rpm), the bacterial cells were harvested (5000 × $g$, 5 min) and submitted to the Magigene Biotechnology Co., Ltd (Guangzhou, China) for RNA sequencing. The total RNA of differently treated NyZ480 was extracted using Trizol reagent, and the NEBNext® Ultra IITM Directional RNA Library Prep Kit for Illumina was used to prepare high-quality strand-specific RNA libraries, with which raw sequencing data were generated with the Illumina NovaSeq PE150 platform. After quality control via fastp[52], raw reads were aligned to the reference genome of NyZ480 (GCA_029537255.1) using Bowtie 2[53]. The expression level of genes was quantified by Salmon[54]; DEGs were analyzed with DESeq2[55]; and ClusterProfile[56] was used to conduct enrichment analysis of DEGs.

For plant transcriptomic analysis, two-week-old Arabidopsis was uniformly selected and subjected to different treatments. Seven days later, root samples from differently treated plants were collected. RNA sequencing was performed by Personalbio Technology Co., Ltd (Shanghai, China). Experimental procedures, including isolation of total RNA, construction of sequencing libraries, and generation of raw data, were performed as described for NyZ480. Then, low-quality reads were filtered by fastp[52], and the retained reads were mapped to the reference genome of Arabidopsis (GCA_000001735.1) using HISAT2[57]. Read counts of each gene were calculated and compared using HTSeq[58] to indicate the original gene expression, and FPKM was then used to standardize the expression. Generation of DEGs and enrichment analysis were carried out as described above. One of the three control replicates was excluded from data analyzes due to contamination.

## Reverse transcription quantitative PCR

Root tissues of Arabidopsis were collected and ground under continuous cooling with liquid nitrogen. Total RNA was extracted using TotalRNAExtractor (Trizol) (Sangon Biotech, Shanghai, China), which was then used for cDNA synthesis with HiScript II Q RT SuperMix for qPCR (+gDNA wiper) (Vazyme, Nanjing, China). Quantitative PCR assays were conducted on the CFX96 real-time PCR detection system (Bio-Rad) with TB Green® Premix Ex TaqTM II (Tli RNaseH Plus) (Takara). *AtTUB*[59] was used as an internal control to normalize transcript levels.

## Antimicrobial assays of simple coumarins on NyZ480 and its mutants

Bacterial cells were cultured and harvested as described above. Then, bacterial suspensions of NyZ480 and its mutants were serially diluted with PBS (pH 7.4), and 10 µL of each diluted bacterial suspension was spotted on 1/6 TSA agar plates containing 1 mM simple coumarin. Incubation was subsequently carried out at 30 °C, and bacterial growth was recorded after 24 h.

## Phenotypic characterization of Arabidopsis plants

Arabidopsis plants were imaged under natural light or 365 nm UV light, and each image included a scale bar. The average fluorescence intensity, the primary root length, and the lateral root count of the Arabidopsis plants were calculated using ImageJ software. The fresh weight of plants was recorded with an analytical balance. Measurement of plant total chlorophyll was conducted as previously reported[13]. Briefly, leaf tissues from three individual plants were pooled and weighed. Samples were mixed with 1 mL DMSO and incubated at 65 °C with shaking. After 60 min, the plant tissues became transparent, and the chlorophyll was completely extracted. An aliquot of 200 µL of each extract was transferred to a 96-well microtiter plate, and absorbance was measured at 652 nm on a multimode microplate reader (Spark, Tecan). Total chlorophyll concentration was calculated with the formula: $\text{Chlorophyll} = \frac{\text{Abs}_{652}\text{nm}}{34.5} * \frac{\text{Volume}_{\text{DMSO}}}{\text{Input weight}}$.

## Arabidopsis iron concentration measurement

Arabidopsis shoots were collected, dried at 65 °C for 48 h, and digested with HNO$_3$. Then, ICP-MS (NexION2000G) was used to analyze iron concentrations.

## Analysis of catabolic genes in environmental isolates

Comprehensive information of "Environmental" and "Host-associated" bacterial isolates was retrieved from the IMG/M database[38], and their assembly accessions were gathered for downloading bacterial

genomes using NCBI Datasets[60], and a total of 51,445 genomes were obtained for further analysis. Amino acid sequences of eight XenAs were aligned by MUSCLE[61] and used to construct a profile Hidden Markov Model (pHMM) with the hmmbuild program of HMMER[62]. The obtained pHMM was used to search for potential XenA homologs in the 51,445 bacterial genomes using the hmmsearch program of HMMER (E-value was set as 1e-50). Amino acid sequences of the downstream catabolic enzymes CouC and MhpBCDEFRT were also queried against the downloaded bacterial genomes using Diamond[63] (identity > 40%, coverage > 70%). The organization of coumarin catabolic genes was analyzed by Cblaster[64].

## Quantification and statistical analysis

HPLC (Waters e2695 Separation Module) was used to analyze the degradation of simple coumarins by NyZ480. Absorption spectra of coumarin, esculetin, fraxetin, and scopoletin were measured at 275 nm, 346 nm, 338 nm, and 344 nm, respectively, with a photodiode array detector (PDA, Waters 2998). A gradient elution was performed at a flow rate of 0.8 mL/min using ultrapure water containing 0.1% formic acid (vol/vol) (solvent A) and acetonitrile (solvent B) as elution solvents. The elution gradient was set as follows: 0–1 min, 5% B; 1–21 min, 5%–95% B; 21–25 min, 95%–5% B. The Agilent 1290 Infinity II/ 6545 QTOF MS system was used for the identification of the intermediates that were derived from the transformation of simple coumarins by the heterologously expressed XenAs. Parent compounds and the intermediates produced by spontaneous hydrolysis were analyzed in negative mode, and the enzymatically produced transient intermediates were analyzed in positive mode. The mass range was set as 50–500 $m/z$. The Agilent 1260/6470 QQQ MS system was used for the quantification of simple coumarins secreted by Arabidopsis plants. Negative mode was used for esculetin, fraxetin, and scopoletin, and positive mode was used for coumarin. Acquisition parameters of MS for different simple coumarins were adjusted using different concentrations of standard substrates (Supplementary Table 5). Elution solvents for LC-MS analyzes were the same as those used for HPLC, and the elution program was modified as follows: 0–1 min, 5% B; 1–21 min, 5%–95% B; 21–31 min, 95% B; 31–32 min, 95%-5% B; 32–42 min, 5% B. The flow rate was 0.3 mL/min. A C18 reversed-phase column (Agilent ZORBAX SB-C18, 5 μm, 4.6 × 250 mm) was used throughout the HPLC and LC-MS analyzes, and the column temperature was 30 °C.

All data visualization was conducted via the online platform Chiplot (https://www.chiplot.online/). Statistical tests were performed using GraphPad Prism 8.0, and details are provided in the figure legends and corresponding method sections.

## Reporting summary

Further information on research design is available in the Nature Portfolio Reporting Summary linked to this article.

## Data availability

Transcriptome data of NyZ480 and Arabidopsis plants have been deposited at NCBI under the BioProject accession PRJNA951349 and PRJNA1204056, respectively. Source data are provided with this paper.

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

## Acknowledgements

We thank Dr. Tao Lu (Zhejiang University of Technology) for sharing experimental materials. We also thank Prof. Zheng Yuan (Shanghai Jiao Tong University), Yazhou Meng, and Xiaoyan Xu (from Prof. Yuan's lab) for their assistance with the Arabidopsis-related experiments. Further, we thank Prof. Qingqiu Gong (Shanghai Jiao Tong University) for valuable discussions during the initial conception of the project. This work was supported by the Fundamental and Interdisciplinary Disciplines Breakthrough Plan of the Ministry of Education of China (JYB2025XDXM906) and the Project of State Key Laboratory of Microbial Metabolism, Shanghai Jiao Tong University (SKLMMCX25_03), both awarded to N.Y.Z.

## Author contributions

Y.C.G and N.Y.Z conceived the research and designed the experiments. Y.C.G performed all molecular and biochemical experiments. Y.C.G and P.P.P conducted bioinformatic analyses. Y.C.G, G.Y., and N.Y.Z drafted

the manuscript. All authors contributed to revising the manuscript and approved the final version.

## Competing interests

The authors declare no competing interests.
