## [Transparent Peer Review file · Nature Communications]

Rhizobacteria opportunistically boost colonization and impair plant fitness by degrading plant-derived coumarins under iron deficiency

Corresponding Author: Professor Ning-Yi Zhou

Version 0:

Reviewer comments:

Reviewer #1

(Remarks to the Author)

In this manuscript, Gu et al. present an insightful study on the interaction between rhizobacteria and plant-derived coumarins under iron-deficient conditions. The authors provide compelling evidence that *Pseudomonas* strain NyZ480 degrades simple coumarins, a crucial class of plant secondary metabolites, leading to opportunistic bacterial colonization and impaired plant fitness. The study identifies a previously overlooked microbial strategy in which rhizobacteria degrade plant-secreted coumarins, thereby limiting plant access to iron and restricting growth. By effectively linking bacterial xenA-mediated coumarin degradation to plant stress under iron deficiency, the authors establish a "vicious cycle" detrimental to the host plant. This work represents a significant contribution to the field of plant-microbe interactions. Overall, this study is highly intriguing, and the following comments are intended to further strengthen the manuscript and enhance its impact:

1. The manuscript discusses transcriptional reprogramming in *Arabidopsis thaliana* following bacterial colonization. However, additional validation of key gene expression changes using qPCR would reinforce the findings.
2. The study convincingly demonstrates that NyZ480 degrades coumarins. However, it would be beneficial to discuss the broader ecological implications, such as whether other plant species that rely on coumarins for iron acquisition experience similar bacterial interference.
3. Given that xenA is widespread among environmental bacteria, it would be valuable to discuss whether this phenomenon commonly occurs in agricultural settings. Addressing this question could enhance the study's broader significance.
4. Chlorophyll measurements suggest a worsening of iron deficiency but do not directly confirm iron uptake inhibition. Quantifying iron levels in plant tissues using ICP-MS or similar methods is strongly recommended.
5. While the study primarily focuses on the bacterial aspects of plant-microbe interactions, it would be worthwhile to explore whether *A. thaliana* employs compensatory mechanisms to counteract bacterial coumarin degradation. For instance, does the plant secrete alternative compounds to recruit beneficial microorganisms and mitigate iron deficiency stress?
6. Figure 6a is too small to read clearly. It is recommended to reformat it for improved readability.
7. Reference formatting: Some citations (e.g., ref51) are inconsistent. Ensure that a uniform citation style is maintained throughout the manuscript.

Reviewer #2

(Remarks to the Author)

This manuscript reports interactions between a rhizosphere-derived *Pseudomonas* isolate (NyZ480) that can catabolise coumarins, important iron-mobilising specialised metabolites secreted by *Arabidopsis* roots under Fe deficiency. The main claim of this manuscript is that coumarin degradation by NyZ480 allows this bacterium to proliferate on iron-stressed roots, out-compete the host for iron and ultimately negatively impact plant growth. This claim suggests an scenario in which rhizosphere microbes opportunistically exploit stress-induced plant metabolites to the detriment of the host.

The topic is timely and potentially important for our understanding of plant-microbe interactions. However, the current dataset does not yet support the central causal chain that links coumarin secretion during Fe deficiency and an increased

proliferation of NyZ480 specifically under these conditions with sufficient rigour or generality for Nature Communications.

-Key controls are missing from multiple experiments, quantitative measurements and corresponding statistical tests are also lacking at points, most conclusions rely on a single bacterial isolate from a non-Arabidopsis host, and several figure panels lack essential information. In addition, the manuscript is poorly written; even the abstract contains grammatical errors and awkward phrasing, and a thorough language edit is essential.

-NyZ480 was isolated from *Plantago asiatica*, not *Arabidopsis*. To demonstrate that the phenomenon is not specific to this one isolate, the authors could for example screen an *Arabidopsis* root-derived strain collection for coumarin degradation, or obtain at least one additional, phylogenetically distinct isolate that shows similar phenotypes. Without these data, it is premature to claim a broad “overlooked incident in the rhizosphere”.

-Figure 1 does not exclude the possibility that lower coumarin levels reflect reduced secretion rather than bacterial degradation; measuring plant growth, chlorophyll concentration and siderophore production would help disentangle these factors.

-Experiments in Figs 1 and 3 lack critical controls. To demonstrate a causal link between coumarin secretion and pathogenicity, experiments in Figures 1 and 3 must include *Arabidopsis* coumarin deficient mutants (e.g., *f6'h1*), chemical complementation with exogenous coumarins, and proper bacterial controls.

-Statements about preferential root colonisation on iron-deficient plates (lines 200–203) are unsupported by quantitative data; fluorescence intensity measurements or CFU counts are required.

-The assertion (lines 216–218) that NyZ480 reduces plant growth regardless of iron supply appears to contradict the idea that coumarin secretion is the key driver of pathogenicity and should be clarified with additional controls.

-Figure 2a requires a no-bacterium control to confirm coumarin stability under the assay conditions, noting that fraxetin is unstable at neutral to alkaline pH without iron.

-Labels beneath the box-and-whisker plots in Figure 3d and information on biological replicates are required.

-Large numbers of DEGs detected when bacterial colonisation is reportedly minimal (lines 252–254) suggest either an underestimation of bacterial load or indirect stress effects; re-quantification of colonisation and presentation of TPM/CPM values for coumarin-biosynthetic and iron-homeostasis genes are needed.

-All bacterial mutants must be shown to grow like wild-type on standard carbon sources *in vitro* and under Fe-sufficient plant conditions *in planta*. In my opinion, quantitative colonisation data and inclusion of these essential controls and fitness tests are critical.

-The bioinformatic survey of *xenA* prevalence would be more convincing if supported by qPCR or metagenomic profiling of root microbiomes from wild-type and *f6'h1* plants grown in iron-replete versus iron-deficient (e.g., calcareous) soils. The analyses presented in the manuscript at best show that these genes are present in a variety of environmental microbes not specifically enriched in roots or rhizosphere of Fe-stressed plants.

-Chlorophyll reported as mg g^{-1} fresh weight represents concentration, not content; the terminology should be corrected.

-Methods that follow established protocols (e.g. chlorophyll extraction) must cite the original publications.

-Figure legends—particularly for supplementary data—need more detail so that experiments are fully interpretable without referring to the main text.

-The authors should discuss why NyZ480 catabolises coumarin and fraxetin but not scopoletin or esculetin, linking this observation to known degradation chemistry.

-I would advise the authors to replace “secondary metabolites” with the now standard term “specialised metabolites”.

-The claim that negative plant outcomes from microbial catabolism of root exudates are “rarely reported” is overstated. Fungal pathogens such as *Alternaria helianthi* and *Botrytis cinerea* have long been shown to degrade coumarins (Tal & Robeson 1986; El Oirdi et al. 2010).

-lines 421-423: Only in certain conditions some of these coumarins can reduce iron. Paffrath et al 2024, has done a very informative analysis on this.

Reviewer #3

(Remarks to the Author)

In this manuscript, Gu et al. explore the effects of bacterial degradation of plant iron-solubilizing metabolites (coumarins) on plant growth under iron limitation. They first demonstrate coumarin degradation by a bacterial isolate (*Pseudomonas* NyZ480) in pure culture and then conduct a series of *in planta* experiments with *Arabidopsis thaliana* showing that the

presence of WT NyZ480 leads to growth defects in iron limited plants, but the presence of coumarin-degradation null mutants does not. A final set of analyses document the presence of coumarin degradation genes in bacterial genomes from diverse environments.

There is growing recognition of the importance of metabolite exchanges in the rhizosphere and coumarins have emerged as especially important and tractable systems for studying these interactions. To date, most studies on coumarins have focused on their role in (1) iron solubilization and stimulation of iron-limited plant growth in bacteria-free systems (2) indirect stimulation of iron-limited plant growth through interactions between coumarins and rhizosphere bacteria or (3) suppression and sculpting of root microbiomes through antibiotic effects of coumarins. The data in this paper offer a new perspective on the ways that coumarins might shape microbial communities and should be of great interest to readers from a variety of fields. In general, the experiments are well done, and the results are for the most part convincing. In particular, the use of LC-MS and mutants to demonstrate coumarin degradation and the gene complements needed is very strong. However, the current iteration of the manuscript does not provide sufficient evidence for two important conclusions about the nature of the in planta experiments. First, although the manuscript concludes that bacterial catabolism is at work in planta and that this limits bacterial colonization, it lacks a clear demonstration that coumarin catabolism rather than coumarin detoxification is at work. Second, it does not explicitly link bacterial colonization and coumarin degradation to plant iron starvation. In addition, while the manuscript is well organized and the ideas are presented in an easy-to-follow order, the writing itself could be improved throughout. Similarly, the figures are mostly of good quality, but some are difficult to interpret due to a lack of labeling or appropriate sizing etc.

Major Comments:

Using qPCR as a metric for bacterial presence and growth, the plant colonization experiments show that WT *Pseudomonas* NyZ480 colonizes iron deplete but not replete plants and that mutant *Pseudomonas* NyZ480 lacking coumarin degradation genes fail to colonize iron deplete plants. In addition, when colonization does occur it leads to decreased plant health via a variety of metrics (fresh weight, root length, chlorophyll). The manuscript takes these data as evidence that this bacterium catabolizes coumarins and that this is required for colonization and growth (due to use of coumarins as a carbon source) and harms the plant. The manuscript also suggests a futile cycle whereby iron-limited plants produce ever more coumarins but due to bacterial coumarin degradation, never relieve iron limitation. This is not an unreasonable interpretation, but it is not the only interpretation, and more work is needed to rule out other explanations. An alternative interpretation is that bacterial presence is generally harmful to the plant and that NyZ480 primarily degrades coumarins as a detoxification mechanism, which does indeed lead to worse plant outcomes under low Fe due partly to coumarin degradation and partly to the stimulation of bacterial colonization by low Fe. I suggest employing the arsenal of tools available for this system including: CFU counts, killed bacterial controls, coumarin null plants, and direct Fe measurements, to address this ambiguity as it pertains to the three topics below.

Interpretation of differences in colonization between low/high Fe conditions: The manuscript currently interprets this as evidence that NyZ480 requires coumarins as a substrate for growth. Why can't NyZ480 use other plant carbon sources as a substrate? How can the alternative possibility that iron stress stimulates bacterial colonization be ruled out? I am also concerned (see below) about the use of qPCR to quantify colonization. CFU counting coupled with experiments under high and low Fe with coumarin-null plants (F6'H1) would easily resolve this issue.

Catabolism vs detoxification in mutants: The manuscript leans heavily on the idea that mutants fail to colonize due to the need for coumarin catabolism. However, it seems equally likely that mutant bacteria are not successful because they lack coumarin degradation genes, which serve to detoxify coumarins. Figure S8 shows mutant growth on TSA plates and looks like a toxicity assay where mutant bacteria are killed by coumarins, rather than one where they fail to catabolize coumarins: TSA should be a good enough carbon source for most bacteria. The manuscript does raise the issue of coumarin detoxification but mostly dismisses it and lines 307-312 interpret the supplemental results as entirely driven by catabolism which I do not think is supported by the data.

Bacterial effects on plant health: The data show decreased plant health in the presence of bacteria in low and high iron conditions regardless of colonization – suggesting bacterial presence, not colonization, is what matters. The manuscript raises the possibility of general bacterially-induced plant stress, especially given the broad plant transcriptomic response to bacterial presence (lines 213-229) but the main thesis of the manuscript is still that the effect seen here is about coumarin degradation and iron starvation. Without a direct measurement of plant iron content, I think that it is difficult to tie the results to the effects of coumarin degradation/iron starvation and this claim needs to be either proven or softened. In addition to measurement of plant Fe, the use of coumarin-null plants as well as heat-killed bacterial additions would help to tease apart these different effects and rule out the possibility that in addition to stimulating plant coumarin production, iron stress also has separate effects on bacterial colonization that are unrelated to coumarins.

Comments on Figures:

Figure 1A: The differences in this figure are very hard to see.

Figure 1B/C. Please explain the reasoning behind adding Fe(II) in an oxic environment where it will be rapidly oxidized to Fe(III).

Figure 2A. Coumarins can degrade over time. Although the time span is relatively short (6 hours), an abiotic degradation control would help here.

Figure 2B. The observed bacterial growth on coumarins is extremely low (max OD= 0.04) and the "control" treatment is a

purely abiotic control. To ensure that this is not just background bacterial growth an additional control with bacteria and without coumarins is needed here. As the authors have previously published on this topic, a reference/discussion of their own previous work showing use of coumarins as a sole carbon source would also help.

Figure 2D. This figure is very hard to interpret. Do the colors correspond to a time course? If so, please label. In addition, the reaction mechanisms are not clear in the current figure, and it is also not clear what structures correspond to the masses shown. These data are presented in the supplement (Fig. S3). I suggest that this figure be clarified/streamlined in some way, possibly by showing a more explicit example with just one coumarin and referencing the supplement for the other compounds.

Figure 3B. The use of qPCR to quantify bacteria does not provide insight into their viability as DNA can stick around long after cells are dead. What is the interpretation of the flat DNA trend across time in the sFe conditions? Are the cells dead and the inoculum DNA is just sticking around? In addition, as the data are normalized to total DNA, which presumably includes plant DNA, how can we be sure that these data do not reflect differences in plant growth? CFU counting would provide much more insight into what is happening. Also, in most plant-microbe experiments bacteria can live on root exuded carbon, why should this not be the case for this organism? How can we be sure that this is about coumarin catabolism vs say Fe starvation stimulating microbial root colonization? The use of a coumarin null plant line (F6'H1) would help with this.

Figure 3C. This figure is very hard to interpret as the plants are not visible in brightfield.

Figure 3D. These data seem inconsistent with those in 3B. In 3B it appears that bacteria do not colonize roots in sFe condition. Yet, in 3D, bacterial presence leads to decreased growth regardless of condition, what is the explanation? A heat-killed control alongside CFU counts of live bacteria would help tease apart the effects of colonization vs a general response to bacteria that harms the plant.

Figure 5B. How can we be sure that the failure of mutant NyZ480 to colonize is due to a need for coumarin catabolism vs coumarin toxicity?

Figure 5C. As with 3C, it is hard to interpret this figure.

Figure 5D. Please label the treatments. These data stand in contrast to those in 3D where the presence of bacteria is harmful regardless of their colonization ability. Based on the framework established in the manuscript, coumarin degradation null bacteria grown on iron limited plants should roughly be the same as WT bacteria grown on Fe replete plants: the difference suggests the importance of coumarin detoxification.

Figure 6. This figure is too small to be readable (especially part A), please make it larger.

Reviewer #4

(Remarks to the Author)

Version 1:

Reviewer comments:

Reviewer #1

(Remarks to the Author)

The authors have adequately addressed all of my previous comments. I have no further comments on the revised manuscript.

Reviewer #2

(Remarks to the Author)

I appreciate the thorough and serious effort the authors have put into revising the manuscript. The revision is clearly substantial and goes well beyond cosmetic changes, and many of the key concerns raised previously have been addressed with new data and clearer analyses, which I acknowledge positively.

First, replacing qPCR based estimates of bacterial abundance with CFU counts significantly improves the rigor of the colonization claims and removes an important methodological weakness. Second, the inclusion of the f6'h1 mutant is a strong addition, as it directly tests the dependence of the observed phenotypes on simple coumarins and makes the causal link much more convincing. Third, the addition of direct plant iron measurements provides an essential connection between microbial activity and plant physiological outcome, which was missing before and is critical for the overall narrative. Fourth, the isolation and characterization of a second strain showing similar behavior strengthens the argument that the findings are not an idiosyncratic feature of a single isolate, even if the taxonomic scope remains limited.

Where I remain unsatisfied is in relation to three main points:

1. The absence of a control using metabolically inactive bacteria still leaves some ambiguity as to whether all observed effects truly depend on active catabolism rather than presence or perception of bacteria, and this point is not fully resolved by the arguments provided.
2. The manuscript still does not directly disentangle catabolism for carbon acquisition from detoxification as the primary driver in planta, and this distinction remains more inferred than demonstrated, which is a conceptual weakness given how central this claim is.
3. while the second isolate is helpful, it is very closely related, and thus does not yet convincingly address the broader generality of the proposed mechanism across more diverse rhizosphere bacteria.

Overall, I think the authors have done a good job and the manuscript is clearly much stronger now. My recommendation would be that the study is close to being acceptable, but still slightly weak mechanistically. In particular, further clarification could come from experiments that more directly demonstrate in planta coumarin catabolism, for example by tracking coumarin derived carbon into bacterial biomass, by measuring specific breakdown products in the rhizosphere during colonization, or by using bacterial mutants that separate detoxification capacity from growth on coumarins as a carbon source. Alternatively, if such experiments are beyond the scope of the current revision, the conclusions should be framed more cautiously to reflect that the current data support a strong correlation and functional dependence, but not yet an unambiguous demonstration of a clear mechanism of nutritional exploitation.

Reviewer #3

(Remarks to the Author)

The authors have done an impressive amount of additional work which greatly strengthens the manuscript and addresses the majority of my initial concerns. My most major comments were that ICP-MS analysis of plant iron status and the use of coumarin null plants were needed to definitively link coumarin degradation and iron sufficiency/deficiency. The authors have provided a nice demonstration of changes in iron content and have used coumarin null plants to link their phenotypes to coumarins. I also suggested that the authors use CFU counts instead of qPCR to test root colonization, again they have provided the appropriate experiments.

I do have one remaining comment (below) about missing controls for microscopy experiment. However, as these data are not crucial for the paper they could be moved to the supplement without the need for further experiments. If the data are going to remain in the main text, it will be important to address this comment. Beyond this my concerns can mostly be addressed with modification of the text rather than additional experiments.

This first pertains to the question of catabolism and detoxification. The MS still emphasizes catabolism, but the data suggest that detoxification is most likely at work, at least some of the time. I agree with the authors that the differences in WT bacterial colonization between iron deficient and iron sufficient Arabidopsis strongly supports the hypothesis of catabolism. However, this does not fully rule out detoxification and an experiment directly demonstrating coumarin catabolism in planta would be needed to prove this one way or the other. This type of experiment would be extremely difficult, and I am not suggesting that the authors attempt this. Instead, I suggest that the possibility of detoxification be introduced earlier in the manuscript, that some of the claims of catabolism be qualified, and that they provide more discussion of the possibilities for detoxification and catabolism in different circumstances.

Also, in general, while the trends observed in the paper are strong, the MS makes no mention of how things may change in more complex microbial communities in the environment. While NyZ480 may grow robustly as a monoculture this might shake out very differently if other microbes were around, the authors should provide a small discussion of this complexity and contextualize some of their claims accordingly.

Finally, the writing of the text is much improved over the previous version but still requires further editing before it will be of publication quality. A few examples are pointed out below, each is minor but as a whole, they detract from the MS.

Small comments:

Line 26: I think it is hard to say that the bacterium is "overproliferating", I would focus this on the results.

Line 32: "to enhance colonization and impair plant fitness" I think this is too strong. While I agree that this is the outcome, the "purpose" from the bacterial standpoint is ambiguous. I suggest something like "enhancing colonization and impairing plant fitness."

Line 58: Into the rhizosphere

Line 64: "Although it still lacks explicit knowledge regarding the composition..." This sentence is a bit confusing.

Line 68: Daphnetin, with an a

71-72: The logic is not clear here

97-98: "An intensive plant-microbe arms race has thus been revealed". I found this a bit too sweeping of a statement.

Line 129: versatily eliminate? I am not sure what this means, this could be clarified.

Line 140: Compared to the control group

Line 187: Arsenal implies multiple weapons. To me, this is just one among many.

Line 230: Induced the mildest influences = had the mildest effects?

Line 285: On the plant's

Line 286: Physiological levels, plural

Line 287: Please cite PMID: 24246380

Line 294: grew limitedly = growth of this strain was limited?

Line 301: Regarding?

Line 360: More declined = decreased?

Line 377: While bacterial use of coumarins seems likely, there is no direct evidence for this so this claim should be softened.

Line 408-409: This sentence needs a bit of clarification

Line 438: As above, I think it is hard to say that the bacterium is "overproliferating", I would focus this on the results rather than editorial comments about it. Also, this sentence is a bit confusing and needs to be revised.

Lines 558-561: I don't think there is strong evidence to support this statement. To prove that this is the case it would be necessary to demonstrate that bacteria are growing on coumarins rather than other exuded carbon in planta, (which would be extremely difficult to do). However, without such an experiment, it remains unclear whether bacteria are detoxifying or catabolizing coumarins.

Figure comments:

Overall the figure captions need to provide more detail on the number of replicates (technical and biological) as well as what the displayed error bars are (SD etc.)

Fig 1B. Small comment: Perhaps just add +10 μ M + NyZ480 to the legend

Fig.1D: Please state the number of replicates used and what the line and shaded region represent.

Fig. 3d: Add "control" or "untreated" or some other designation to indicate the no bacteria treatments

Fig 3A/5A: The CFU counts are hard to see, please decrease the y-axis scale to better match the data (mostly in 5A). Also, the statistics are a little confusing. How were the comparisons that are displayed chosen? Please clarify. The box plots make it hard to track the differences between the different conditions perhaps consider presenting these data as a line graph?

Fig 3B/5B: These experiments require further controls. The authors have used different replicating plasmids with RFP for the WT as opposed to the mutant strains. Were microscopy experiments done using kanamycin/tetracycline/other antibiotics to maintain the plasmid in NyZ480? Otherwise, the signal could be missed due to plasmid loss. In addition, the use of two different plasmids could lead to differences in RFP expression levels. It would be nice to show that RFP is produced at equivalent amounts and is not lost when antibiotics are absent within each strain and/or to show a positive control for the mutant strains in 3B to ensure the difference is not just due to low fluorescence. In addition, in 5B where the microscopy images show no colonization it is important to include a positive control. These experiments are not strictly necessary for the paper as the CFU data provide strong evidence, so if this cannot be improved the data could be moved to the supplement.

Reviewer #4

(Remarks to the Author)

Version 2:

Reviewer comments:

Reviewer #2

(Remarks to the Author)

The author's have addressed all my concerns and I am happy to recommend this manuscript for publication.

Reviewer #3

(Remarks to the Author)

The authors have adequately addressed my concerns and the manuscript is scientifically sound and the claims are reasonable. My only remaining recommendation is that the authors conduct further editing of writing for clarity and the removal of persistent typos and confusing phrasing.

Reviewer #4

(Remarks to the Author)

Point-by-point response letter

Rhizobacteria opportunistically boost colonization and impair plant fitness by degrading plant-derived coumarins under iron deficiency

By Yichao Gu et al.

We sincerely appreciate the reviewers' thoughtful and generally positive evaluation of our work. In the revised manuscript, we have carefully addressed all major and minor concerns and substantially expanded the dataset. In particular, we have now included new data derived from direct measurement of plant iron concentrations by ICP - MS, experiments incorporating the simple coumarin - deficient mutant *f6'h1*, re-quantification of bacterial colonization by CFU counting, isolation and characterization of an additional coumarin-degrading *Pseudomonas* strain (NyZ490) from the *Arabidopsis* rhizosphere, RT - qPCR validation of expression levels of key *Arabidopsis* simple coumarin biosynthesis genes after NyZ480 inoculation, and a bioinformatic survey of Xena occurrence and abundance in metagenomic datasets. We are grateful for the critical and rigorous assessment, which has clearly strengthened both the mechanistic and ecological conclusions of our study.

For transparency, we provide both a clean version of the revised manuscript and a version with all changes highlighted using Track Changes. Below, we respond to each reviewer's comments in a detailed, point-by-point manner. Reviewer comments are shown in black, and our responses are shown in blue. Unless otherwise indicated, line numbers refer to the clean revised main-text document.

Reviewer #1 (Remarks to the Author)

In this manuscript, Gu et al. present an insightful study on the interaction between rhizobacteria and plant-derived coumarins under iron-deficient conditions. The authors provide compelling evidence that *Pseudomonas* strain NyZ480 degrades simple coumarins, a crucial class of plant secondary metabolites, leading to opportunistic bacterial colonization and impaired plant fitness. The study identifies a previously overlooked microbial strategy in which rhizobacteria degrade plant-secreted coumarins, thereby limiting plant access to iron and restricting growth. By effectively linking bacterial *xenA*-mediated coumarin degradation to plant stress under iron deficiency, the authors establish a "vicious cycle" detrimental to the host plant. This work represents a significant contribution to the field of plant-microbe interactions.

We appreciate the reviewer's thorough and insightful review of our manuscript. Your comprehensive understanding of our work and the highly positive assessment you provided are both encouraging and invaluable. To further strengthen the manuscript, we have

42 conducted additional experiments and expanded the discussion in accordance with the
43 reviewers' suggestions. The point-by-point responses to your comments are provided as
follows.

Overall, this study is highly intriguing, and the following comments are intended to further
strengthen the manuscript and enhance its impact:

1. The manuscript discusses transcriptional reprogramming in *Arabidopsis thaliana* following
bacterial colonization. However, additional validation of key gene expression changes using qPCR
would reinforce the findings.

**Thank you for the suggestion. As we mentioned in the “Discussion” of the previous version of**
**our manuscript, *Arabidopsis* transcriptome data indicated that the combined treatment of**
**iron deficiency and NyZ480 inoculation further enhanced the expression of several coumarin**
**biosynthesis genes, including *4cl*, *f6'h1*, *s8h*, and *cyp82c4*, compared to iron deficiency alone**
**(previous Supplementary Fig. 10a, now Supplementary Fig. 11a). Following your advice, we**
**have performed RT-qPCR to quantify the expression levels of these genes, including *4cl*, *f6'h1*,**
***cosy*, *s8h*, and *cyp82c4* in *Arabidopsis* roots across different treatments. The new results (now**
**Supplementary Fig. 11c) are consistent with the RNA-seq data, indicating that NyZ480**
**colonization significantly stimulates the biosynthesis of simple coumarins in iron-deficient**
***Arabidopsis* Col-0. We again thank the reviewer for this valuable suggestion. The inclusion of**
**the RT – qPCR results has further strengthened the conclusions presented in our manuscript.**

2. The study convincingly demonstrates that NyZ480 degrades coumarins. However, it would be
beneficial to discuss the broader ecological implications, such as whether other plant species that
rely on coumarins for iron acquisition experience similar bacterial interference.

**We thank the reviewer for this insightful suggestion. Although the tug-of-war over iron**
**between plant and microbe unfolds through distinct strategies and mechanisms, no precedent**
**exists, to our knowledge, for bacterial degradation of root-secreted coumarins impairing iron**
**uptake in other plant species. Given the conserved role of simple coumarins in iron acquisition**
**across plant taxa, more future field investigations will help to establish the broader ecological**
**significance of this plant-microbe interaction.**

**In light of the phenomena revealed in this study and the current research status, we can**
**primarily highlight the potential significance of this mechanism. Accordingly, we have**
**emphasized in the revision that “Given that iron deficiency challenges global agricultural**
**productivity, greater attention should be directed to the potential disruption of plant iron**
**acquisition by coumarin-degrading rhizobacteria” (lines 568–570). We believe this statement**
**reasonably reflects the implications of our findings while acknowledging the need for further**
**validation.**

3. Given that *xenA* is widespread among environmental bacteria, it would be valuable to discuss
whether this phenomenon commonly occurs in agricultural settings. Addressing this question could
enhance the study's broader significance.

**We thank the reviewer for raising this point. In the revision, we have re-analyzed the dataset**
**retrieved from IMG database and identified 346 bacterial genomes from agriculture-**
**associated environments, such as agricultural land, pasture, paddy field and etc. Among these,**
**over 97% (336/346) harbor *xenA*, and gene redundancy is high as well. However, none of these**
**genomes contain the full genetic determinants for complete catabolism of simple coumarins.**
**These results align with our earlier observations in the previous version (now in lines 392-407,**
**lines 414-416, and lines 419-420), and collectively imply that the plant-microbe interaction**
**revealed in this study may be relatively uncommon in natural settings. That said, we have also**
**isolated an additional coumarin-degrading strain, *Pseudomonas* sp. NyZ490 (its 16S rRNA**
**sequence identity with NyZ480 is 99.07%), which exerts detrimental effects on iron-deficient**
***Arabidopsis Col-0* similar to those of NyZ480 (lines 536-544). Hence, as stated in the**
**manuscript, “it cannot be ruled out that coumarin-degrading bacteria with the potential to**
**disrupt plant iron uptake may reside in natural settings.” (lines 547-548). In summary, based**
**on the current data and results, it is premature to conclude how commonly the plant-microbe**
**interaction revealed in this study occurs in agricultural settings.**

4. Chlorophyll measurements suggest a worsening of iron deficiency but do not directly confirm
iron uptake inhibition. Quantifying iron levels in plant tissues using ICP-MS or similar methods is
strongly recommended.

**We thank the reviewer for suggesting ICP-MS quantification of iron levels in plant tissues. We**
**acknowledge this important recommendation and wish to clarify that we did employ ICP-MS**
**for direct measurement of plant iron concentration. These data were all consolidated and**
**presented in Fig. 5e in the previous version, which certain has resulted in confusion between**
**the narrative sequence of the “Results” section and the data presentation. To enhance**
**readability and coherent presentation, we have recompiled and presented the iron content**
**data of both previously measured *Col-0* and newly added mutant *f6'h1* in the revised Fig. 3d,**
**with the aim of clearly demonstrating the impact of NyZ480 inoculation on plant iron**
**absorption under different conditions.**

5. While the study primarily focuses on the bacterial aspects of plant-microbe interactions, it would
be worthwhile to explore whether *A. thaliana* employs compensatory mechanisms to counteract
bacterial coumarin degradation. For instance, does the plant secrete alternative compounds to recruit
beneficial microorganisms and mitigate iron deficiency stress?

**Thanks for raising this point. Our study primarily focused on the roles of simple coumarins**
**in plant iron acquisition and did not investigate whether *Arabidopsis* compensatorily employs**
**alternative metabolites to facilitate its iron acquisition, directly or indirectly. The reviewer's**

**valuable perspective have prompted our reassessment of published evidences and uncovered**
**these following information: (1) Benzoxazinoids are used as a chelator for plant iron uptake¹,**
**however, only by maize plants but not Arabidopsis; (2) Mugineic acid secreted by gramineous**
**plants is an iron-chelating siderophore that helps plants in iron acquisition²; (3) Arabidopsis-**
**secreting quercetin facilitates iron absorption via reduction³; (4) Kaempferol derived from**
**tomato or grape is capable of chelating Fe(III)⁴, etc. Nevertheless, no studies have documented**
**compensatory enhancement in the secretion of these metabolites when coumarin-mediated**
**iron acquisition is impaired. Hence, continued efforts are required in the future to investigate**
**complementary mechanisms among distinct iron acquisition strategies in plants. Such insights**
**will enable us to further study how plants' exudation influence microbial communities and the**
**resultant feedback effects of microbes on plant hosts.**

6. Figure 6a is too small to read clearly. It is recommended to reformat it for improved readability.

**Thanks for pointing out this issue. We have reformatted the figure accordingly.**

7. Reference formatting: Some citations (e.g., ref51) are inconsistent. Ensure that a uniform citation
style is maintained throughout the manuscript.

**We have thoroughly checked the reference format and made corrections accordingly.**

**Reviewer #2 (Remarks to the Author):**

This manuscript reports interactions between a rhizosphere-derived *Pseudomonas* isolate (NyZ480)
that can catabolise coumarins, important iron-mobilising specialised metabolites secreted by
*Arabidopsis* roots under Fe deficiency. The main claim of this manuscript is that coumarin
degradation by NyZ480 allows this bacterium to proliferate on iron-stressed roots, out-compete the
host for iron and ultimately negatively impact plant growth. This claim suggests an scenario in
which rhizosphere microbes opportunistically exploit stress-induced plant metabolites to the
detriment of the host.

The topic is timely and potentially important for our understanding of plant – microbe interactions.
However, the current dataset does not yet support the central causal chain that links coumarin
secretion during Fe deficiency and an increased proliferation of NyZ480 specifically under these
conditions with sufficient rigour or generality for Nature Communications.

Key controls are missing from multiple experiments, quantitative measurements and corresponding
statistical tests are also lacking at points, most conclusions rely on a single bacterial isolate from a
non-*Arabidopsis* host, and several figure panels lack essential information. In addition, the
manuscript is poorly written; even the abstract contains grammatical errors and awkward phrasing,
and a thorough language edit is essential.

**We sincerely appreciate the reviewer’s thorough understanding of our study's core**
**contributions and the recognition of its timely significance. The reviewer’s rigorous critique,**
**which is essential for maintaining Nature Communications' publication standards has been**
**invaluable in identifying opportunities for refinement. As rightly noted, many aspects in our**
**previous version require strengthening. Accordingly, we have performed a series of additional**
**experiments, including the incorporation of the *Arabidopsis* mutant *f6'h1*; directly**
**measurement of plant iron concentration using ICP-MS; re-evaluation of bacterial**
**colonization by CFU counting; isolation and characterization of a new coumarin-degrader**
**from the *Arabidopsis* rhizosphere, and analysis of *xenA* appearance and abundance in**
**metagenomic datasets, etc. All identified omissions of critical information, grammatical**
**inaccuracies, and problematic phrasing in the manuscript have been thoroughly checked and**
**revised. Regarding points where our interpretation diverges, we have also provided detailed**
**explanations to support our perspective. The point-by-point responses are listed below.**

-NyZ480 was isolated from *Plantago asiatica*, not *Arabidopsis*. To demonstrate that the phenomenon
is not specific to this one isolate, the authors could for exampl screen an *Arabidopsis* root-derived
strain collection for coumarin degradation, or obtain at least one additional, phylogenetically distinct
isolate that shows similar phenotypes. Without these data, it is premature to claim a broad
“overlooked incident in the rhizosphere” .

**To address the reviewer's concern regarding whether the plant-microbe interaction mediated**
**by NyZ480 is an isolated case, we conducted an additional screening for coumarin-degrading**
**strains in the *Arabidopsis* rhizosphere (rhizospheric soil was collected from 28-day-old**

*Arabidopsis thaliana* Col-0 grown in Pindstrup Substrate, a commercial peat-based
compost/soil mix). This effort led to the isolation of *Pseudomonas* sp. NyZ490 (its 16S rRNA
sequence identity with NyZ480 is 99.07%), which is another simple coumarin degrader
(Supplementary Fig. 12a,b). Furthermore, we demonstrated that NyZ490 robustly colonized
the roots of iron-deficient *Arabidopsis* Col-0 and, similarly, impaired plant growth
(Supplementary Fig. 12c-e). Although the newly obtained NyZ490 is of the same genus as
NyZ480, we believe our findings still demonstrate that the plant-microbe interaction revealed
in our study is not specific to just one isolate and may represent a conserved trait within
certain *Pseudomonas* species (lines 544-548). Given that the reviewer might consider the term
“overlooked” too strong, we have replaced it with “unreported” (line 30, line 99).

-Figure 1 does not exclude the possibility that lower coumarin levels reflect reduced secretion rather
than bacterial degradation; measuring plant growth, chlorophyll concentration and siderophore
production would help disentangle these factors.

**First, Fig. 1c (now Fig. 1b) demonstrates near-undetectable levels of simple coumarins in the**
**liquid medium of iron-deficient *Arabidopsis* Col-0 inoculated with NyZ480. We concluded in**
**the previous version that the increased secretion of simple coumarins under iron deficiency**
**was degraded by NyZ480. Indeed, as rightly noted by the reviewer, we cannot completely**
**exclude the possibility that bacterial inoculation directly reduced plant coumarins secretion.**
**However, RNA-seq and newly provided RT-qPCR Data in Supplementary Fig. 11 reveal that**
**NyZ480 inoculation significantly elevated the expression of simple coumarin biosynthesis**
**genes, including *4cl*, *f6'h1*, *cosy*, *s8h*, and *cyp82c4*, compared to iron deficiency alone. These**
**transcriptional evidences strongly indicate that simple coumarin biosynthesis and secretion**
**are increased, upon NyZ480 inoculation. Therefore, the scenario suggested by the reviewer**
**perhaps appears less likely, and our interpretation remains supported by the combined**
**metabolic and transcriptional data.**

-Experiments in Figs 1 and 3 lack critical controls. To demonstrate a causal link between coumarin
secretion and pathogenicity, experiments in Figures 1 and 3 must include *Arabidopsis* coumarin
deficient mutants (e.g., *f6'h1*), chemical complementation with exogenous coumarins, and proper
bacterial controls.

**We agree with the reviewer that additional controls are critically needed in our study. We have**
**now supplemented new experimental data using a simple coumarin synthesis-deficient mutant**
***f6'h1*. The obtained results indicate that *f6'h1* no longer secretes simple coumarins**
**(Supplementary Data1), and NyZ480 fails to establish enhanced colonization on *f6'h1* (Fig.**
**5a,b), as it does on iron-deficient Col-0 (Fig. 3a,b). Together, these newly obtained data provide**
**in planta physiological evidence that the specialized metabolites, simple coumarins, are**
**directly associated with NyZ480's colonization and the resultant damage to plant fitness.**

**Regarding the other suggested controls involving exogenous coumarins supplementation or**

the use of proper bacterial strains, we may not fully appreciate the experimental rationale
behind these specific approaches. In a previous study of Tsai et al., the authors demonstrated
the indispensability of simple coumarins for Arabidopsis iron uptake by supplementing
coumarins biosynthesis mutants with exogenous fraxetin⁵. While we acknowledge that this
experimental paradigm validates *in vitro* significance of simple coumarins in plant iron
acquisition, it is probably unnecessary to replicate identical controls in our system, as the iron-
mobilizing function of simple coumarins has already been well documented, and our
manuscript explicitly references this consensus. As for proper bacterial controls, we maintain
that the NyZ480 mutant strains are the optimal bacterial controls for this study. The
corresponding data (Fig. 3, Fig. 5, Supplementary Fig. 8) demonstrate at the physiological
level that, in contrast to the wild-type strain, Mut-NyZ480 fails to establish enhanced
colonization on iron-deficient Col-0, suggesting a direct association between NyZ480's
catabolic capacity of simple coumarins and its enhanced colonization on Col-0 under iron
deficiency. We are inclined to think that employing other bacterial strains as controls would
introduce confounding variables: such strains might damage or benefit plants through
mechanisms unrelated to simple coumarin degradation or utilization, thereby diverting focus
from our core investigation of metabolite-dependent pathogenesis.

-Statements about preferential root colonisation on iron-deficient plates (lines 200 – 203) are
unsupported by quantitative data; fluorescence intensity measurements or CFU counts are required.

We acknowledge the concerns raised by the reviewer regarding the use of qPCR for
quantifying NyZ480 colonization. In response to your suggestions, we have re-evaluated
bacterial colonization in relevant experiments using CFU counting. The new data are
presented in the revised Fig. 3a and Fig. 5a, showing that NyZ480 displays significantly
enhanced proliferation on iron-deficient Arabidopsis Col-0 ($\log_{10}(\text{CFU}/\text{root})$ value reaching
7.5 at 7 DPI), while also exhibiting limited growth on iron-sufficient Col-0 plants
($\log_{10}(\text{CFU}/\text{root})$ value exceeding 4 at 7 DPI) (lines 196-201). The newly obtained CFU data
underscore the limitation of qPCR in accurately quantifying bacterial colonization. Therefore,
the qPCR data were excluded and replaced with CFU counts.

-The assertion (lines 216 – 218) that NyZ480 reduces plant growth regardless of iron supply appears
to contradict the idea that coumarin secretion is the key driver of pathogenicity and should be
clarified with additional controls.

Thank you for raising this question. But, to some extent, we have a differing view on this
particular point. Actually, we are not intended to claim that “coumarin secretion is the key
driver of pathogenicity” in the manuscript. Instead, the core message conveyed here is that
the secretion of simple coumarins by iron-deficient Arabidopsis Col-0 further exacerbates the
NyZ480-induced damage to plant fitness. (lines 207-223)

Although plant fresh weight, primary root length, and lateral root count are affected by

**NyZ480 inoculation under both iron-deficient and iron-sufficient conditions, total chlorophyll**
**concentration and iron levels are further reduced only in iron-deficient Arabidopsis Col-0. We**
**agree that NyZ480 colonization is generally pathogenic to Arabidopsis regardless of iron**
**condition, but the *worsened* detriment occurs exclusively when simple coumarins are secreted,**
**which is primarily under iron deficiency. Thus, we have added in the revised manuscript: “the**
**worsened bacterial detriment to plants was iron deficiency-dependent” (lines 222-223). We**
**consider the data and associated interpretations in this section sound, and hope our response**
**could adequately alleviate the reviewer's doubts.**

-Figure 2a requires a no-bacterium control to confirm coumarin stability under the assay conditions,
noting that fraxetin is unstable at neutral to alkaline pH without iron.

**We thank the reviewer for raising this valid concern. As suggested, we have now included the**
**requested data in Supplementary Fig. 2. The results indicate that, with the exception of**
**fraxetin, the other three simple coumarins remained stable throughout the 6-hour assay**
**period. Moreover, although fraxetin showed instability, clear evidence of bacterial**
**degradation was still observed (Fig. 1c).**

-Labels beneath the box-and-whisker plots in Figure 3d and information on biological replicates are
required.

**We have supplemented the requested labels and information as suggested in the revised Fig.**
**3a,c,d and Fig. 5a,c.**

-Large numbers of DEGs detected when bacterial colonisation is reportedly minimal (lines 252 -
254) suggest either an underestimation of bacterial load or indirect stress effects; re-quantification
of colonisation and presentation of TPM/CPM values for coumarin-biosynthetic and iron-
homeostasis genes are needed.

**We thank the reviewer for raising this point. After re-quantification of bacterial colonization**
**using CFU counting, we have found a relatively limited growth of NyZ480 on iron-sufficient**
**Arabidopsis Col-0 (Fig. 3a), suggesting there was an underestimation of bacterial load caused**
**by qPCR assays. Therefore, the large numbers of DEGs detected when NyZ480 inoculated on**
**iron-sufficient Arabidopsis Col-0 are associated with bacterial colonization as well, even**
**though the colonization is restricted compared to that on iron-deficient Col-0. Hence, we have**
**added in the manuscript: “even limited colonization of WT-NyZ480 exerted non-negligible**
**effects on Arabidopsis growth.” (lines 250-251).**

**Regarding your request for the TPM/CPM values of genes related to coumarin synthesis and**
**iron homeostasis, we are uncertain of the intent behind this request. We infer that you may**
**wish to understand whether WT-NyZ480 inoculation affects iron uptake or homeostasis in**

iron-sufficient Arabidopsis Col-0. To address this specifically, we have extracted and analyzed
 the expression data for genes involved in simple coumarin biosynthesis and iron homeostasis
 from the RNA-seq dataset. As requested, the expression levels for these key genes in both
 control (iron-sufficient roots: sFe_r1, sFe_r2, sFe_r3) and experimental groups (iron-
 sufficient roots inoculated with NyZ480: sFe480_r1, sFe480_r2, sFe480_r3) are summarized
 below. As demonstrated by these data, the expression of genes involved in simple coumarin
 synthesis and iron homeostasis is not induced upon NyZ480 inoculation under iron-sufficient
 conditions. This finding reinforces our interpretation that the transcriptional changes
 observed in iron-sufficient plants exposed to NyZ480, despite the limited bacterial colonization,
 are not mediated through alterations in plant's iron status or simple coumarins production
 pathway.

Table. R1 Transcriptional level of simple coumarin biosynthesis and iron homeostasis genes in roots of
 Arabidopsis Col-0 under iron sufficiency (sFe_r) and NyZ80 inoculation (sFe480_r).

Gene	Gene_D	sFe_r1	sFe_r2	sFe_r3	sFe480r1	sFe480r2	sFe480r3
PAL1	AT2G37040	290.3346625	174.8718812	207.895261	169.8745725	174.9770822	131.7277988
PAL2	AT3G53260	116.135957	76.66452425	103.2373916	72.90003199	79.64827317	54.77357564
PAL3	AT5G04230	0.079684226	0.116130669	0.109035235	0.086220435	0.082697757	0.133245248
PAL4	AT3G10340	35.80136591	22.26193362	31.50193536	19.55288784	19.65763953	16.33139691
C4H	AT2G30490	392.1132885	297.5303153	294.4229438	284.9283466	284.7172494	268.6758888
C3H	AT2G40890	81.64197421	65.17792236	88.45717469	55.34822266	60.40664206	56.96357164
4CL1	AT1G51680	97.89323954	72.66663263	93.43154722	70.59779118	71.20956555	38.08648462
4CL3	AT1G65060	23.75687656	11.19885531	13.16086519	3.190335241	8.795031536	7.517215117
4CL4	AT3G21230	9.585925004	9.009654654	14.64121809	6.20882769	8.69619707	6.445330707
4CL-like1	AT1G62940	0.061181914	0.118887548	0.047838716	0.052960357	0.158739293	0.15345947
4CL-like2	AT1G20480	5.368210319	6.611825251	6.826163093	5.585590971	5.346127398	6.120367105
4CL-like4	AT1G20500	0	0	0	0	0	0
4CL-like5	AT1G20510	11.02892162	9.638367479	12.24340817	7.240128714	15.94771453	8.212790515
4CL-like6	AT4G19010	7.291048439	8.264566646	7.343918723	6.90296847	5.832729564	6.011255836
4CL-like7	AT4G05160	31.80394849	24.44464294	34.18630198	15.76076605	26.8961849	28.42935581
4CL-like8	AT5G38120	0	0	0	0	0	0
4CL-like9	AT5G63380	11.6815431	10.63879076	12.09305855	7.032389821	11.44252928	11.70743038
HCT	AT5G48930	163.4872185	127.8428023	133.9280479	180.0342753	136.4672176	113.319957
COMT	AT5G54160	160.4355522	135.1659497	171.8130696	91.18177904	128.3092567	92.97602168
CCoAOMT1	AT4G34050	419.8160865	417.3047092	443.6466728	276.7890018	382.0664495	265.1291698
F6'H1	AT3G13610	30.76014391	21.95094803	43.62471128	9.457044443	20.51172989	21.63862035
COSY	AT1G28680	1.147892826	0.557640951	0.872616341	0.745230198	0.463285008	0.906415104
SSH	AT3G12900	1.982060705	0.071324133	12.36963596	0	0	0
CYP82C4	AT4G31940	4.659741182	0.455414994	37.14640799	0.477345603	0.171691072	0
FT	AT2G28160	23.57662805	14.19784391	31.36269849	4.172969097	14.18846308	22.42005782
MYB72	AT1G56160	0.050265025	0	0.510935072	0	0	0
BGLU42	AT5G36890	11.87590924	14.28056622	17.6537748	8.736077722	10.87227621	9.098662167
PDR9	AT3G53480	81.67506936	55.47398749	75.89470006	29.5389666	58.99069734	72.68079547

 -All bacterial mutants must be shown to grow like wild-type on standard carbon sources in vitro and
 under Fe-sufficient plant conditions in planta. In my opinion, quantitative colonisation data and
 inclusion of these essential controls and fitness tests are critical.

 **Thank you for your suggestion. Regarding the growth data on standard carbon sources, we**
 **have assessed the growth of all NyZ480 mutants in LB medium, and results show comparable**
 **growth patterns among all strains (Supplementary Fig. 9), indicating that the knockout of the**
 **catabolic genes disrupts the strain's ability to degrade simple coumarins without affecting its**
 **growth on standard carbon sources. (lines 354-356). Regarding the colonization experiments**
 **on iron-sufficient plants, we selected NyZ480Δ*xenAsΔmhpB* as the representative of all**
 **NyZ480 mutants, and inoculated it on iron-sufficient Arabidopsis. Results show that**
 **NyZ480Δ*xenAsΔmhpB* exhibits a more declined colonization than WT-NyZ480 on both Col-**

**0 and *f6'h1* Arabidopsis (Fig. 3a,b, Fig. 5a,b), and caused ameliorated damage to plant fitness**
 **compared to that of WT-NyZ480 (Fig. 3c, Fig. 5c). These collectively indicates “the coumarin**
 **catabolic genes in NyZ480 are essential not only for the opportunistic, enhanced colonization**
 **on iron-deficient Col-0 but also for the limited growth of NyZ480 on Arabidopsis, irrespective**
 **of plant genotype or iron status.” (lines 366-368), and “the growth impairment of Arabidopsis**
 **was linked to the extent of bacterial colonization, which was most pronounced in iron-deficient**
 **Col-0 following WT-NyZ480 inoculation.” (lines 380-382).**

-The bioinformatic survey of *xenA* prevalence would be more convincing if supported by qPCR or
 metagenomic profiling of root microbiomes from wild-type and *f6'h1* plants grown in iron-replete
 versus iron-deficient (e.g., calcareous) soils. The analyses presented in the manuscript at best show
 that these genes are present in a variety of environmental microbes not specifically enriched in
 roots or rhizosphere of Fe-stressed plants.

**We thank the reviewer for the constructive comments. It is indeed necessary to support the**
 **prevalence of *XenA* in the environment with more analytical data. Fortunately, the article**
 **published by Stringlis et al.⁶ provides sequencing data from metagenomes of rhizosphere soils**
 **of Arabidopsis Col-0 and its mutants *myb72* and *f6'h1*. We therefore utilized these publicly**
 **available data to analyze the prevalence of *xenA*.**

**First, we found that the *XenA* homologs are universally present across all sample groups**
 **collected by Stringlis et al. (hmmsearch, e-value threshold of $< e^{-40}$), with at least 20 counts of**
 ***XenA* homologous sequences in each group (see figure below), consistent with our analysis in**
 **the previous version. However, regarding whether *XenA* is enriched in the rhizosphere of iron-**
 **deficient Col-0, our analysis showed no statistically significant differences in the abundance**
 **(TPM values) of *XenA* homologs among all sample groups (see figure below).**

**Figure. R1 Count and abundance of *XenA* homologs in soil samples of different Arabidopsis**

**lines.** Metagenomic data from Stringlis et al.'s paper were downloaded from NCBI database, and
were processed as previously described⁷. HMMER⁸ was used to search for XenA homologs
(hmmsearch, e-value threshold of < e-40), and XenA abundance (TPM value) was calculated using
CoverM⁹. The relative abundance of XenA in each sample was represented by the sum of the
abundances of all XenA homologs present in that sample. Each group comprises three replicates.
dFe: iron deficiency; sFe: iron sufficiency; *f6'h1*: simple coumarin biosynthesis mutant; *myb72*:
MYB transcription factor mutant.

**We would like to further emphasize that comparing XenA abundance in rhizospheric soils of**
**differently treated Arabidopsis lines (Col-0 and *f6'h1*) at a single time point may not**
**adequately assess whether XenA is enriched by coumarin-secreting plants. A more**
**appropriate approach would be involving analysis of XenA abundance in time-course**
**metagenomic samples from the iron-deficient Col-0 rhizosphere. Such a longitudinal design**
**could capture an increase in XenA abundance over time, as plants continually secrete simple**
**coumarins under iron deficiency, thereby providing more compelling evidence for enrichment.**
**However, such time-resolved metagenomic data are not currently available. As the re-analyzed**
**dataset from Stringlis et al. did not show significant XenA enrichment in iron-deficient Col-0,**
**we cannot yet conclude whether XenA is actually selectively enriched in the rhizosphere under**
**iron stress.**

**However, as noted above, the isolation of an additional coumarin degrader, NyZ490, exhibiting**
**the same phenotype on Arabidopsis as NyZ480 (Supplementary Fig. 12, lines 536-544),**
**suggests that the underlying trait may not be unique among rhizobacteria. Although it**
**remains unclear whether the ecological interaction described in this study occurs widely in**
**natural settings, the phenomenon observed in this study still warrants further attention.**

-Chlorophyll reported as mg g⁻¹ fresh weight represents concentration, not content; the
terminology should be corrected.

**We are grateful to the reviewer for bringing this point to our attention. Terminology has been**
**changed as suggested.**

-Methods that follow established protocols (e.g. chlorophyll extraction) must cite the original
publications.

**We thank the reviewer for pointing this out. The original reference regarding chlorophyll**
**extraction has been cited as requested (line 749).**

-Figure legends—particularly for supplementary data—need more detail so that experiments are
fully interpretable without referring to the main text.

**As suggested, we have carefully reviewed and revised all figure legends to ensure they contain**
**sufficient experimental detail. The supplementary figure legends, in particular, have been**
**expanded to make the experiments fully interpretable as standalone items. We hope the**
**updated versions meet the reviewer's expectations.**

-The authors should discuss why NyZ480 catabolises coumarin and fraxetin but not scopoletin or
esculetin, linking this observation to known degradation chemistry.

**We thank the reviewer for raising this important point regarding the substrate specificity of**
**NyZ480. As the reviewer rightly points out, the differing catabolism of these structurally**
**similar compounds is likely attributable to the subtle variations in their ring substituents.**
**These differences could critically impact the catalytic efficiency of downstream enzymes (e.g.,**
**those encoded by *couC* or the *mhp* cluster). Specifically, distinct substituents may not be**
**optimally accommodated within the enzyme's active site due to steric hindrance, or they may**
**fail to form essential interactions (e.g., hydrogen bonds) with key amino acid residues in the**
**active enzyme pocket. Given that our study primarily explores the biological consequences of**
**this catabolism in the plant-host context, a detailed enzymatic investigation was not pursued**
**in this study.**

-I would advise the authors to replace “secondary metabolites” with the now standard term
“specialised metabolites” .

**We thank the reviewer for this constructive comment. As suggested, we have replaced the term**
**"secondary metabolites" with the now standard term "specialized metabolites" across the**
**entire text.**

-The claim that negative plant outcomes from microbial catabolism of root exudates are “rarely
reported” is overstated. Fungal pathogens such as *Alternaria helianthi* and *Botrytis cinerea* have
long been shown to degrade coumarins (Tal & Robeson 1986; El Oirdi et al. 2010).

**We thank the reviewer for this insightful comment and for highlighting the important work**
**on coumarin-degrading fungal pathogens. We agree that the degradation of coumarins by**
**pathogenic fungi is well-documented. In our manuscript, however, the specific statement in**
**the abstract refers to bacterial catabolism and its detrimental effects, not microbial catabolism**
**in a broader sense. To the best of our knowledge, and within the context of bacterial**
**colonization, such negative outcomes for the plant host remain less frequently reported. We**
**have therefore retained the original phrasing " However, detrimental effects on plants arising**
**from bacterial colonization that exploits plant-derived metabolites are rarely documented."**
**(lines 18-19) to accurately reflect this distinction. We appreciate the reviewer's expertise,**
**which has allowed us to clarify this point.**

-lines 421-423: Only in certain conditions some of these coumarins can reduce iron. Paffrath et al
2024, has done a very informative analysis on this.

**We thank the reviewer for pointing this out. The reference by Paffrath et al. (2024) was indeed**
**cited elsewhere in our original manuscript, but we inadvertently omitted it from this specific**
**statement. We have now inserted the citation (line 453) and have slightly refined the text to**
**more accurately reflect the findings of that study.**

**Reviewer #3 (Remarks to the Author):**

In this manuscript, Gu et al. explore the effects of bacterial degradation of plant iron-solubilizing
metabolites (coumarins) on plant growth under iron limitation. They first demonstrate coumarin
degradation by a bacterial isolate (*Pseudomonas* NyZ480) in pure culture and then conduct a series
of in planta experiments with *Arabidopsis thaliana* showing that the presence of WT NyZ480 leads
to growth defects in iron limited plants, but the presence of coumarin-degradation null mutants does
not. A final set of analyses document the presence of coumarin degradation genes in bacterial
genomes from diverse environments.

There is growing recognition of the importance of metabolite exchanges in the rhizosphere and
coumarins have emerged as especially important and tractable systems for studying these
interactions. To date, most studies on coumarins have focused on their role in (1) iron solubilization
and stimulation of iron-limited plant growth in bacteria-free systems (2) indirect stimulation of iron-
limited plant growth through interactions between coumarins and rhizosphere bacteria or (3)
suppression and sculpting of root microbiomes through antibiotic effects of coumarins. The data in
this paper offer a new perspective on the ways that coumarins might shape microbial communities
and should be of great interest to readers from a variety of fields. In general, the experiments are
well done, and the results are for the most part convincing. In particular, the use of LC-MS and
mutants to demonstrate coumarin degradation and the gene complements needed is very strong.
However, the current iteration of the manuscript does not provide sufficient evidence for two
important conclusions about the nature of the in planta experiments. First, although the manuscript
concludes that bacterial catabolism is at work in planta and that this limits bacterial colonization, it
lacks a clear demonstration that coumarin catabolism rather than coumarin detoxification is at work.
Second, it does not explicitly link bacterial colonization and coumarin degradation to plant iron
starvation. In addition, while the manuscript is well organized and the ideas are presented in an easy-
to-follow order, the writing itself could be improved throughout. Similarly, the figures are mostly
of good quality, but some are difficult to interpret due to a lack of labeling or appropriate sizing etc.

**We are deeply grateful for the reviewer's thorough evaluation of our manuscript and the**
**insightful comments. We particularly appreciate the reviewer's profound knowledge on the**
**coumarin research field and the recognition that our work provides an important new**
**perspective for this area.**

**In response to the reviewer's valuable suggestions, we have performed additional experiments:**
**introducing control groups using coumarin-null plants (*fb'h1*), re-evaluating bacterial**
**colonization through CFU counting as suggested and measuring plant iron level via ICP-MS.**
**Hopefully, the new data would enhance credibility to our conclusions. Also, we have**
**thoroughly polished the writing throughout the manuscript and carefully revised all figures**
**and figure legends to address the issues identified regarding unclear labeling and**
**inappropriate sizing, ensuring accurate presentation and clear visualization. We believe these**
**substantive additions and revisions have significantly enhanced the quality of our manuscript.**
**Once again, we sincerely thank the reviewer for the time and expertise you have dedicated to**
**improving our manuscript. We hope the revised version meets with your approval.**

Major Comments:

Using qPCR as a metric for bacterial presence and growth, the plant colonization experiments show
that WT *Pseudomonas* NyZ480 colonizes iron deplete but not replete plants and that mutant
*Pseudomonas* NyZ480 lacking coumarin degradation genes fail to colonize iron deplete plants. In
addition, when colonization does occur it leads to decreased plant health via a variety of metrics
(fresh weight, root length, chlorophyll). The manuscript takes these data as evidence that this
bacterium catabolizes coumarins and that this is required for colonization and growth (due to use of
coumarins as a carbon source) and harms the plant. The manuscripts also suggests a futile cycle
whereby iron-limited plants produce ever more coumarins but due to bacterial coumarin degradation,
never relieve iron limitation. This is not an unreasonable interpretation, but it is not the only
interpretation, and more work is needed to rule out other explanations. An alternative interpretation
is that bacterial presence is generally harmful to the plant and that NyZ480 primarily degrades
coumarins as a detoxification mechanism, which does indeed lead to worse plant outcomes under
low Fe due partly to coumarin degradation and partly to the stimulation of bacterial colonization by
low Fe. I suggest employing the arsenal of tools available for this system including: CFU counts,
killed bacterial controls, coumarin null plants, and direct Fe measurements, to address this
ambiguity as it pertains to the three topics below.

**We thank the reviewer for the accurate summary of our work and for rightly pointing out that**
**alternative interpretations of our experimental results exist. To address these potential**
**ambiguities and solidify our conclusions, we have employed the recommended arsenal of tools**
**(CFU counting, coumarin null plants(*f6'h1*), ICP-MS measurement of Fe, etc.) and performed**
**additional experiments. Hopefully, the detailed responses listed below are adequate.**

Interpretation of differences in colonization between low/high Fe conditions: The manuscript
currently interprets this as evidence that NyZ480 requires coumarins as a substrate for growth. Why
can't NyZ480 use other plant carbon sources as a substrate? How can the alternative possibility
that iron stress stimulates bacterial colonization be ruled out? I am also concerned (see below) about
the use of qPCR to quantify colonization. CFU counting coupled with experiments under high and
low Fe with coumarin-null plants (F6' H1) would easily resolve this issue.

**We acknowledge the reviewer's point that qPCR may not be the most accurate method for**
**quantifying bacterial colonization, and we agree that our initial interpretation was**
**preliminary. In the revised manuscript, we have replaced the qPCR data with CFU counts,**
**and conducted additional experiments using coumarin-null plants (*f6'h1*) under both iron-**
**sufficient and iron-deficient conditions.**

**As for Arabidopsis Col-0, CFU counting indicated that WT-NyZ480 exhibited a minor**
**colonization on iron-sufficient roots (Fig. 3a), suggesting that the bacterium can utilize other**
**plant-derived carbon sources than coumarins, for limited growth. However, this is negligible**
**compared to the enhanced colonization of WT-NyZ480 observed in iron-deficient Col-0. As we**

state in the revised manuscript, “these observations demonstrated that opportunistic
colonization by NyZ480 was strongly promoted by the iron-deficient status of the host plant.”
(lines 204-206).

In addition, for *f6'h1*, which is impaired in simple coumarin biosynthesis, the CFU counts of
WT-NyZ480 did not exhibit significant increase under either iron conditions compared to the
levels seen in iron-deficient Col-0 plants (Fig. 3a, Fig. 5a). “These observations strongly
supported a direct correlation between the enhanced WT-NyZ480 colonization and the
production of simple coumarins by Arabidopsis under iron deficiency.” (lines 298-300).

Catabolism vs detoxification in mutants: The manuscript leans heavily on the idea that mutants fail
to colonize due to the need for coumarin catabolism. However, it seems equally likely that mutant
bacteria are not successful because they lack coumarin degradation genes, which serve to detoxify
coumarins. Figure S8 shows mutant growth on TSA plates and looks like a toxicity assay where
mutant bacteria are killed by coumarins, rather than one where they fail to catabolize coumarins:
TSA should be a good enough carbon source for most bacteria. The manuscript does raise the issue
of coumarin detoxification but mostly dismisses it and lines 307-312 interpret the supplemental
results as entirely driven by catabolism which I do not think is supported by the data.

We appreciate the reviewer's insightful comment, which rightly points out that both
catabolism and detoxification are supported by our data. Indeed, these two processes are
intrinsically linked and functionally important.

The observed growth recession of the mutant NyZ480 on TSA plates directly demonstrates
that the redundant *xenA* gene in NyZ480 is essential for detoxifying esculetin and scopoletin.
This detoxification capability allows the strain to utilize carbon and energy sources from TSA
even under the bacteriostatic stress imposed by these compounds.

However, the detoxification capability empowered by *xenA* only allows the strain to utilize
other carbon and energy sources for growth in the presence of antibacterial simple coumarins,
but does not enable it to grow using coumarin and fraxetin, which are significantly secreted
by iron-deficient Arabidopsis Col-0. Therefore, in the context of the enhanced colonization of
WT-NyZ480 on iron-deficient Col-0, which significantly secretes simple coumarins,
detoxification ability alone is inadequate. The strain must also be capable of catabolizing
accessible simple coumarins, such as coumarin and fraxetin, to thrive in this environment.
This requires both *xenA* and the presence of downstream genes necessary for catabolizing
simple coumarins

In summary, while *xenA*-mediated detoxification of antimicrobial esculetin and scopoletin is
clearly important, the catabolic capacity of the strain appears to play a more critical role in
promoting enhanced colonization under iron-deficient conditions. That said, we fully
acknowledge the importance of the detoxification capacity in NyZ480's interaction with
simple coumarins and have accordingly expanded our discussion of this aspect in the revised

**manuscript to address the reviewer's concerns (lines 338-351).**

Bacterial effects on plant health: The data show decreased plant health in the presence of bacteria
in low and high iron conditions regardless of colonization – suggesting bacterial presence, not
colonization, is what matters. The manuscript raises the possibility of general bacterially-induced
plant stress, especially given the broad plant transcriptomic response to bacterial presence (lines
213-229) but the main thesis of the manuscript is still that the effect seen here is about coumarin
degradation and iron starvation. Without a direct measurement of plant iron content, I think that it
is difficult to tie the results to the effects of coumarin degradation/iron starvation and this claim
needs to be either proven or softened. In addition to measurement of plant Fe, the use of coumarin-
null plants as well as heat-killed bacterial additions would help to tease apart these different effects
and rule out the possibility that in addition to stimulating plant coumarin production, iron stress also
has separate effects on bacterial colonization that are unrelated to coumarins.

**In our initial submission, we reported that WT-NyZ480 inoculation led to impaired plant**
**growth in both iron-sufficient and iron-deficient Arabidopsis Col-0 roots. By using qPCR**
**quantification, we observed significant colonization only under iron-deficient conditions,**
**which has raised the reviewer's concern that the detrimental effect of NyZ480 on plant growth**
**might occur independently of colonization. However, in this revised manuscript, we have re-**
**evaluated the bacterial biomass using CFU counts, showing that WT-NyZ480 exhibited slight**
**growth even on iron-sufficient Col-0 roots (Fig. 3a) (lines 196-201). This updated result**
**indicates that the compromised plant fitness and the transcriptomic changes of iron-sufficient**
**Col-0 are still likely associated with bacterial colonization.**

**In response to the reviewer's suggestions regarding iron concentration measurement, we have**
**employed ICP-MS to directly quantify iron levels of plants. The obtained results demonstrate**
**that (Fig. 3d) shoot iron concentration of iron-deficient Col-0 further decreased after WT-**
**NyZ480 inoculation, confirming that "WT-NyZ480 indeed interfered with iron absorption in**
**plants under iron stress, and the worsened bacterial detriment to plants was iron deficiency-**
**dependent." (lines 221-223).**

**By using the coumarin-null mutant *fc6'h1*, we have confirmed that the enhanced root**
**colonization by WT-NyZ480 under iron deficiency is specifically linked to plant-derived**
**simple coumarins, rather than other iron stress-induced effects. This conclusion is supported**
**by the absence of significantly enhanced colonization of WT-NyZ480 on *fc6'h1* (Fig. 5a,b)**
**compared to that on iron-deficient Col-0.**

**Due to an erroneous estimation of NyZ480 colonization on iron-sufficient Arabidopsis roots**
**by qPCR data, the reviewer might perceive that the compromised plant growth of iron-**
**sufficient Col-0 is unrelated to bacterial colonization. Hence, the reviewer brought out the**
**suggestion of incorporating a heat-killed bacteria control. However, through CFU counting,**
**we have found that WT-NyZ480 also grew to a limited extent and colonized Col-0 roots under**
**iron-sufficient conditions. This may have led to the impaired plant fitness (except for total**

chlorophyll and shoot iron concentration) and the large numbers of DEGs observed in our
experimental results. Therefore, the aforementioned phenomena are indeed related to the
relatively limited bacterial colonization. For this reason, we did not introduce heat-killed
bacteria as a control to exclude factors causing plant growth impairment and transcriptional
reprogramming that are unrelated to bacterial colonization.

Comments on Figures:

Figure 1A: The differences in this figure are very hard to see.

**We apologize for the lack of clarity in the previous Fig. 1a. As this figure was intended to show**
**plant chlorosis under different culture conditions, magnified views of the shoots are presented**
**to allow clear observation of the differential leaf chlorosis phenotypes under various iron**
**culture conditions. Hopefully, this modification meets the reviewer's standard.**

Figure 1B/C. Please explain the reasoning behind adding Fe(II) in an oxic environment where it will
be rapidly oxidized to Fe(III).

**We thank the reviewer for raising this point. The standard MS medium contains Fe²⁺ that is**
**chelated by Na₂EDTA, which is also an ingredient of MS medium. This Fe-EDTA complex is**
**designed to remain stable under oxic conditions, which effectively prevents oxidation and**
**precipitation, thereby maintaining iron bioavailability.**

Figure 2A. Coumarins can degrade over time. Although the time span is relatively short (6 hours),
an abiotic degradation control would help here.

**We thank the reviewer for raising this valid concern regarding the potential spontaneous**
**degradation of coumarins. To address this, we performed a set of control experiments without**
**bacterial inoculation. The results, now included as Supplementary Fig. 2, demonstrate that**
**most simple coumarins remained stable over the 6-hour incubation period throughout the**
**experiment procedure, with the exception of fraxetin. Even though fraxetin is unstable, clear**
**evidence of bacterial degradation was still observed (Fig. 1c).**

Figure 2B. The observed bacterial growth on coumarins is extremely low (max OD= 0.04) and the
“control” treatment is a purely abiotic control. To ensure that this is not just background bacterial
growth an additional control with bacteria and without coumarins is needed here. As the authors
have previous published on this topic, a reference/discussion of their own previous work showing
use of coumarins as a sole carbon source would also help.

**We appreciate the reviewer's comment on the low OD₆₀₀ values. The growth on coumarin has**
**been previously validated in our work titled "Elucidation of the coumarin degradation by**

*Pseudomonas* sp. strain NyZ480"¹⁰. However, in the revised manuscript, no-substrate controls
have still been included so as to relieve the reviewer's concern that low OD values may not
indicate bacterial growth. The obtained data (Fig. 1d) confirmed that the measured growth is
attributable to the substrate, because no background growth was recorded in each no-
substrate control.

Figure 2D. This figure is very hard to interpret. Do the colors correspond to a time course? If so,
please label. In addition, the reaction mechanisms are not clear in the current figure, and it is also
not clear what structures correspond to the masses shown. These data are presented in the
supplement (Fig. S3). I suggest that this figure be clarified/streamlined in some way, possibly by
showing a more explicit example with just one coumarin and referencing the supplement for the
other compounds.

**We thank the reviewer for the constructive feedback on the previous Fig. 2d. We agree that**
**the original presentation could be improved for clarity. The colors in the curves indeed**
**correspond to a reaction time course, and we have now clearly labeled this in the revised**
**figures (now Fig. 2a). Regarding figure reorganization, we fully agree with the reviewer's**
**suggestion to streamline the figures (previous Fig. 2d and Fig. S3). Accordingly, we have**
**plotted the new Fig. 2 in the main text, which focuses exclusively on the enzymatic reaction of**
**fraxetin by the eight redundant XenA enzymes. This new figure integrates the Extracted iron**
**chromatograms (EICs) and the corresponding mass spectrometry data. Moreover, the**
**analogous reaction data for esculetin and scopoletin have been moved to the new**
**Supplementary Fig. 4 to maintain a focused narrative in the main text while keeping the**
**complete dataset available. We hope these revisions are satisfactory.**

Figure 3B. The use of qPCR to quantify bacteria does not provide insight into their viability as DNA
can stick around long after cells are dead. What is the interpretation of the flat DNA trend across
time in the sFe conditions? Are the cells dead and the inoculum DNA is just sticking around? In
addition, as the data are normalized to total DNA, which presumably includes plant DNA, how can
we be sure that these data do not reflect differences in plant growth? CFU counting would provide
much more insight into what is happening. Also, in most plant-microbe experiments bacteria can
live on root exuded carbon, why should this not be the case for this organism? How can we be sure
that this is about coumarin catabolism vs say Fe starvation stimulating microbial root colonization?
The use of a coumarin null plant line (F6' H1) would help with this.

**We thank the reviewer for these critical questions, which have significantly strengthened our**
**study. We fully agree that qPCR has limitations in assessing viable bacterial colonization, and**
**the use of total DNA also cannot exclude the interference caused by plant growth. To address**
**the concerns in this regard, we have replaced all bacterial colonization data with CFU counts**
**in the revised manuscript. The new data confirmed that WT-NyZ480 established a low but**
**detectable level of colonization on iron-sufficient Col-0 plants (Fig. 3a), consistent with the**
**reviewer's comment that bacteria can utilize various root exudates for growth. However, this**

colonization is negligible compared to the thriving growth of WT-NyZ480 on iron-deficient
Col-0 roots, suggesting that “opportunistic colonization by NyZ480 was strongly promoted by
the iron-deficient status of the host plant.” (lines 205-206).

Nevertheless, as the reviewer questioned, whether this enhanced colonization of NyZ480 is
attributed to coumarin catabolism or other iron starvation-related factors? We have
performed a key experiment using the coumarin-null *f6'h1*. The obtained results are definitive:
Growth of WT-NyZ480 on both iron-sufficient and iron-deficient *f6'h1* was similar, but
significantly less than its growth on iron-deficient Col-0. This in planta physiological evidence
indicates “a direct correlation between the enhanced WT-NyZ480 colonization and the
production of simple coumarins by Arabidopsis under iron deficiency.” (lines 298-300).

Collectively, the CFU data and the *f6'h1* experiment robustly demonstrate that the enhanced
colonization of NyZ480 on iron-deficient Col plants is not a general response to iron starvation
but is specifically driven by the bacterial catabolism and detoxification of host-derived simple
coumarins. We have revised the manuscript accordingly, hopefully, these new data fully
address the reviewer's concerns.

Figure 3C. This figure is very hard to interpret as the plants are not visible in brightfield.

We apologize for the lack of clarity in the original Fig. 3c. To ensure the Arabidopsis roots are
clearly visible under bright field, we reduced the magnification and have now provided
reacquired images in Fig. 3b. Hopefully, those images meet the reviewer's standard.

Figure 3D. These data seem inconsistent with those in 3B. In 3B it appears that bacteria do not
colonize roots in sFe condition. Yet, in 3D, bacterial presence leads to decreased growth regardless
of condition, what is the explanation? A heat-killed control alongside CFU counts of live bacteria
would help tease apart the effects of colonization vs a general response to bacteria that harms the
plant.

The reviewer is absolutely reasonable to question how the presence of WT-NyZ480 could
impair plant growth under iron-sufficient condition if it fails to colonize the roots. This
apparent discrepancy prompted us to re-evaluate the colonization data with CFU counting as
the reviewer suggested. In the revised manuscript, we have found that WT-NyZ480 established
a low, but detectable colonization on iron-sufficient Col-0 plants, though it is substantially less
than that on iron-deficient Col-0 plants, while the Mut-NyZ480 showed an even more declined
growth on Arabidopsis, regardless of plant genotype (Col-0 or *f6'h1*) or iron condition (Fig.
3a, Fig. 5a).

As for the plant growth after bacterial inoculation, “Because the colonization capacity of Mut-
NyZ480 was compromised, the growth defects observed in Col-0 and *f6'h1* after WT-NyZ480
treatment were partially alleviated when Mut-NyZ480 was inoculated instead” (lines 372-374).

**When we correlate the plant growth phenotype with the revised bacterial colonization data, it**
**could be concluded that the growth impairment observed in iron-sufficient plants inoculated**
**with WT-NyZ480 is likely attributable to the slight bacterial colonization. Hence, “WT-**
**NyZ480 colonization generally impaired Arabidopsis fitness, and that iron-deficient culture**
**conditions particularly boosted enhanced proliferation and exacerbated the detrimental**
**effects of this bacterium.” (lines 212-214).**

**Since the new data has already provided clear evidence that bacterial colonization is directly**
**associated with the deteriorated plant fitness, we decided to omit the additional heat-killed**
**bacteria control.**

**Figure 5B. How we be sure that the failure of mutant NyZ480 to colonize is due to a need for**
**coumarin catabolism vs coumarin toxicity?**

**We suppose the reviewer is curious about whether it is the lack of coumarin catabolism or**
**detoxification that causes the failure of Mut-NyZ480 to colonize Arabidopsis. As shown in Fig.**
**3a and Fig. 3b, Mut-NyZ480 shows barely increased growth on iron-sufficient Col-0 (the**
**log₁₀(CFU/root) value all remained below 4 at 7 DPI), which does not secrete esculetin or**
**scopoletin—compounds with antimicrobial activity against NyZ480. This finding clearly**
**indicates that the declined colonization of Mut-NyZ480 is not caused by its inability to detoxify**
**simple coumarins. In addition, we observed that WT-NyZ480 exhibits limited growth**
**(log₁₀(CFU/root) value exceeding 4 at 7 DPI) on Arabidopsis plants that do not secrete simple**
**coumarins (i.e. iron-sufficient Col-0, iron-sufficient or iron-deficient *f6'h1*). This suggests that**
**WT-NyZ480 may utilize other root-secreted compounds to support its limited growth. In**
**contrast, the mutant strain Mut-NyZ480 is unable to effectively colonize any of these**
**Arabidopsis plants. Therefore, as stated in the manuscript, we propose that “the coumarin**
**catabolic genes in NyZ480 are essential not only for the opportunistic, enhanced colonization**
**on iron-deficient Col-0 but also for the limited growth of NyZ480 on Arabidopsis, irrespective**
**of plant genotype or iron status.”, and that “these catabolic genes may also act on other, non-**
**specific SMs secreted by Arabidopsis roots, thereby enabling this limited bacterial**
**colonization”.** (lines 366–371)

**In summary, based on the combined evidence, we conclude that the failure of Mut-NyZ480 to**
**thrive and significantly colonize iron-deficient Arabidopsis Col-0 results from a loss of**
**catabolic function toward simple coumarins, rather than a defect in detoxification.**

**Figure 5C. As with 3C, it is hard to interpret this figure.**

**We apologize for the lack of clarity in this figure. We have now provided reacquired images in**
**Fig. 5b. Hopefully, those images meet the reviewer’s standard.**

Figure 5D. Please label the treatments. These data stand in contrast to those in 3D where the presence
of bacteria is harmful regardless of their colonization ability. Based on the framework established
in the manuscript, coumarin degradation null bacteria grown on iron limited plants should roughly
be the same as WT bacteria grown on Fe replete plants: the difference suggests the importance of
coumarin detoxification.

**Regarding figure labeling, we have now clearly labeled all treatments in the revised Fig. 3a,c,d,**
**Fig. 5a,c as suggested.**

**Discrepancy between previous Fig. 5d and Fig. 3d: The reviewer correctly points out a seeming**
**contradiction. We agree that our original conclusion that the detrimental effect on plant**
**growth was independent of bacterial colonization was inaccurate. This error stemmed from a**
**limitation of the qPCR method used initially to quantify colonization. By using CFU counting,**
**we revealed a critical difference that WT-NyZ480 exhibited low but detectable colonization on**
**iron-sufficient Col-0 plants (Fig. 3a, lines 196-201), while the mutant-NyZ480 barely grew on**
**Col-0 and *f6'h1*, regardless of iron conditions (Fig. 3a, Fig. 5a, lines 359-362). This explains**
**why the impaired plant growth was to some extent relieved when inoculated with mutant**
**NyZ480 compared to WT-NyZ480 (Fig. 3c and Fig. 5c, lines 372-377), as bacterial colonization**
**observed in this study generally harms plant fitness and mutant-NyZ480 barely colonizes**
**Arabidopsis, irrespective of genotype or culture condition.**

**Regarding the point you raised about coumarin detoxification, our latest data may not fully**
**support this view. This is because Mut-NyZ480 does not show significant colonization on the**
**roots of either Col-0 or *f6'h1*, regardless of iron conditions. Given that *f6'h1* does not secrete**
**simple coumarins, the inability of Mut-NyZ480 to colonize Arabidopsis roots is likely**
**unrelated to its loss of detoxification capacity of simple coumarins. In the revised manuscript,**
**we have made an alternative speculation that “the coumarin catabolic genes in NyZ480 are**
**essential not only for the opportunistic over-colonization on iron-deficient Col-0, but also for**
**the limited growth of NyZ480 on Arabidopsis, irrespective of genotype or iron status. It was**
**thus reasonable to speculate that these catabolic genes may also act on other non-specific SMs**
**secreted by Arabidopsis roots, enabling the limited bacterial colonization.” (lines 366-371).**

Figure 6. This figure is too small to be readable (especially part A), please make it larger.

**We have revised Fig. 6 as suggested to improve its readability.**

**Referenec:**

- 1. Hu, L. *et al.* Plant iron acquisition strategy exploited by an insect herbivore. *Science* **361**, 694–697
(2018).
- 2. Yamagata, A. *et al.* Uptake mechanism of iron-phytosiderophore from the soil based on the structure
of yellow stripe transporter. *Nat Commun* **13**, 7180 (2022).
- 3. Xiao, L., Luo, G., Tang, Y. & Yao, P. Quercetin and iron metabolism: What we know and what we
need to know. *Food and Chemical Toxicology* **114**, 190–203 (2018).
- 4. Dimitrić Marković, J. M., Amić, D., Lučić, B. & Marković, Z. S. Oxidation of kaempferol and its
iron(III) complex by DPPH radicals: spectroscopic and theoretical study. *Monatsh Chem* **145**, 557–
563 (2014).
- 5. Tsai, H.-H. *et al.* Scopoletin 8-hydroxylase-mediated fraxetin production is crucial for iron
mobilization. *Plant Physiol.* **177**, 194–207 (2018).
- 6. Stringlis, I. A. *et al.* MYB72-dependent coumarin exudation shapes root microbiome assembly to
promote plant health. *Proc. Natl. Acad. Sci. U.S.A.* **115**, E5213–E5222 (2018).
- 7. Pan, P., Gu, Y., Li, T., Zhou, N.-Y. & Xu, Y. Deciphering the triclosan degradation mechanism in
*Sphingomonas* sp. strain YL-JM2C: Implications for wastewater treatment and marine resources. *J.*
*Hazard. Mater.* **478**, 135511 (2024).
- 8. Finn, R. D., Clements, J. & Eddy, S. R. HMMER web server: interactive sequence similarity
searching. *Nucleic Acids Research* **39**, W29–W37 (2011).
- 9. Aroney, S. T. N. *et al.* CoverM: read alignment statistics for metagenomics. *Bioinformatics* **41**,
btaf147 (2025).
- 10. Gu, Y., Li, T., Yin, C.-F. & Zhou, N.-Y. Elucidation of the coumarin degradation by *Pseudomonas* sp.
strain NyZ480. *J. Hazard. Mater.* **457**, 131802 (2023).

Point-by-point response letter

**Rhizobacteria opportunistically boost colonization and impair plant**
**fitness by degrading plant-derived coumarins under iron deficiency**

By Yichao Gu et al.

**We thank the editor and reviewers for their favorable assessment of our revised manuscript**
**and also for their rigorous, constructive comments, which have substantially strengthened the**
**paper. In this second revised version, we have replaced conclusions that lacked direct**
**experimental support with more cautious interpretations based on our current results.**
**Regarding the suggested experiments that are difficult to perform within the scope of this**
**study, we have accordingly refined the relevant statements in the manuscript to be more**
**circumspect and better aligned with the available evidence. For ease of review, we are**
**submitting both a clean version of the manuscript and a version with all changes highlighted.**
**Below, we respond to the reviewers' comments point by point. Our responses appear in blue,**
**and line numbers refer to the contents in the clean version of the manuscript.**

**Reviewer #1 (Remarks to the Author):**

The authors have adequately addressed all of my previous comments. I have no further comments
on the revised manuscript.

**We thank the reviewer for the positive feedback and for confirming that all issues have been**
**addressed.**

**Reviewer #2 (Remarks to the Author):**

I appreciate the thorough and serious effort the authors have put into revising the manuscript. The
revision is clearly substantial and goes well beyond cosmetic changes, and many of the key concerns
raised previously have been addressed with new data and clearer analyses, which I acknowledge
positively.

First, replacing qPCR based estimates of bacterial abundance with CFU counts significantly
improves the rigor of the colonization claims and removes an important methodological weakness.
Second, the inclusion of the *f6' h1* mutant is a strong addition, as it directly tests the dependence
of the observed phenotypes on simple coumarins and makes the causal link much more convincing.
Third, the addition of direct plant iron measurements provides an essential connection between
microbial activity and plant physiological outcome, which was missing before and is critical for the
overall narrative. Fourth, the isolation and characterization of a second strain showing similar
behavior strengthens the argument that the findings are not an idiosyncratic feature of a single isolate,
even if the taxonomic scope remains limited.

**We are grateful for the reviewer's careful review of our revised manuscript and for**
**acknowledging the efforts we have made. We sincerely appreciate the valuable feedback the**
**reviewer provided earlier in the first-round review, which has significantly enhanced the**
**scientific rigor and persuasiveness of our work.**

Where I remain unsatisfied is in relation to three main points:

1. The absence of a control using metabolically inactive bacteria still leaves some ambiguity as to
whether all observed effects truly depend on active catabolism rather than presence or perception of
bacteria, and this point is not fully resolved by the arguments provided.

**We thank the reviewer for this important point. We believe the catabolically defective mutant**
**NyZ480 Δ 8xenAs Δ mhpB already serves as a highly specific control that directly links**
**coumarin degradation to both enhanced root colonization and the subsequent negative effects**
**on plant fitness. This mutant is no longer able to use simple coumarins as carbon sources and**
**severely impaired in detoxifying esculetin and scopoletin (Supplementary Fig. 9). Critically,**
**unlike the wild-type NyZ480, it shows no enhanced root colonization at 7 DPI and causes no**
**impairment in the growth or iron absorption of either Arabidopsis Col-0 or *f6'h1*, regardless**
**of plant iron status (Fig. 3b,c and Fig. 5b).**

**Thus, the clear absence of impaired plant phenotypes with this living but catabolically inactive**
**mutant demonstrates that active coumarin degradation, rather than mere bacterial presence**
**or plant perception of bacteria, is responsible for the observed detrimental effects. We believe**
**this clarification directly addresses the reviewer's concern.**

2. The manuscript still does not directly disentangle catabolism for carbon acquisition from
detoxification as the primary driver in planta, and this distinction remains more inferred than
demonstrated, which is a conceptual weakness given how central this claim is.

**Indeed, our data point to the involvement of both catabolism and detoxification of simple**
**coumarins. We posit that these two processes are intrinsically linked during the colonization**
**of iron-deficient Arabidopsis Col-0 by NyZ480.**

**Regarding the detoxification aspect, the presence of redundant *xenA* genes in NyZ480 (Fig. 1e)**
**ensures the degradation of different simple coumarins (Fig. 2 and Supplementary Fig. 4),**
**thereby detoxifying these antimicrobial phytochemicals (Supplementary Fig. 9c). This**
**capability likely allows *xenA*-harboring bacteria to survive in simple coumarin-enriched**
**environment, potentially by utilizing alternative carbon sources. Conversely, the catabolism**
**of simple coumarins (e.g. coumarin and fraxetin) directly fuels bacterial growth, and this**
**enhanced bacterial colonization consequently impacts the plant host. Thus, in the specific**
**context of iron-deficient Arabidopsis Col-0, which secretes significant amounts of simple**
**coumarins, we propose that catabolism serves as the primary driver of NyZ480 colonization.**

**As rightly noted by the reviewer, the distinction between catabolism and detoxification during**
**NyZ480 colonization is mostly inferred from our *ex planta* results. Given that the initiating**
**gene *xenA* is required for both detoxification and catabolism of simple coumarins, it is**
**technically challenging to generate mutant strains that genetically uncouple these functions**
**for further experimentation. Consequently, we have revised the relevant sections of the**
**manuscript to present the mechanistic discussions with greater caution, refraining from**
**causally defining catabolism as the sole driver for NyZ480 colonization and raising the**
**contributory role of detoxification. We hope this revision satisfactorily addresses the**
**reviewer's concern.**

**The revisions are made as follows:**

**(1) In order to incorporate the contributory role of detoxification, revisions are made in**
**several parts of the manuscript:**

**“Hence, we reveal an unreported rhizospheric phenomenon in which microorganisms**
**opportunistically utilize and detoxify host-secreted specialized metabolites under stress**
**conditions, enhancing colonization and impairing plant fitness.” (Lines 29-32).**

**We changed “degrade and utilize” to “utilize and detoxify”**

**“Our findings establish that bacterial utilization and detoxification of plant-derived simple**
**coumarins drive enhanced bacterial colonization and worsened growth impairment in iron-**
**stressed Arabidopsis.” (Lines 90-93).**

**We changed “degradation and utilization” to “utilization and detoxification”.**

**“Given that these plant-derived compounds are widely present in the rhizosphere and exhibit**
**antimicrobial activity, the redundant *xenA* genes harbored by the rhizosphere-isolated**

NyZ480 likely represent an adaptive detoxification strategy, evolved in the ongoing tug-of-war
between rhizospheric microorganisms and their host plants.” (Lines 183-187).

We changed “may be considered as genetic arsenals” to “likely represent an adaptive
detoxification strategy”, which would allow to introduce the concept of detoxification earlier
and better emphasize its potential functional significance.

“Having demonstrated, using the Arabidopsis mutant *f6’h1*, that host production of simple
coumarins is indispensable for the enhanced colonization of WT-NyZ480 under iron deficiency,
we employed NyZ480 mutants to further investigate whether the bacterial_utilization and
detoxification capacity against these phytochemicals underpinned its proliferation on iron-
stressed plant roots.” (Lines 316-320).

We changed “degradation and utilization” to “utilization and detoxification”.

“Collectively, *xenA* redundancy equips NyZ480 with the dual capacity to utilize certain simple
coumarins as carbon sources and to detoxify others’ antimicrobial effects.” (Lines 348-350).

We changed “withstand antimicrobial effects of others” to “detoxify others’ antimicrobial
activity”.

(2) In order to qualify the definitive conclusion, the following sentences are revised:

“....., demonstrating that the disrupted plant iron acquisition was likely caused by WT-
NyZ480’s degradation of simple coumarins.”

We replaced “indeed” with “likely”, and “WT-NyZ480’ s degradation and utilization” with
“WT-NyZ480’ s degradation” (Lines 375-376). These changes qualify the conclusion and
thereby temper the inference regarding the primary role of coumarin catabolism (utilization).

“Since NyZ480 not only degrades but also grows on simple coumarins, this continual
degradation and utilization massively deplete the iron-mobilizing coumarins, likely further
compromising plant uptake of the limited iron sources.” (Lines 470-473).

We added “likely” to qualify the conclusion.

“This proposed framework explains why” (Line 477).

We added “proposed” to moderate the inferred conclusion.

“In particular, the iron deficiency-induced enhanced colonization of NyZ480 further perturbs
plant iron acquisition through continuous degradation of simple coumarins,” (Line 520-
522).

We changed “through the degradation and utilization of simple coumarins” to “through
continuous degradation of simple coumarins”, which reduces emphasis on catabolism
(utilization) as the main driver underlying NyZ480’s enhanced colonization and subsequent
negative impact on plants.

We changed “It is the sustained bacterial growth on simple coumarins that perpetuates the
elimination of these iron-mobilizing SMs constantly secreted by plants, thereby establishing a

“vicious circle” that progressively compromises plant health.” to “Here, we propose that the
sustained degradation of simple coumarins by NyZ480 perpetuates the elimination of these
iron-mobilizing SMs from the rhizosphere. This persistent removal may create a feedback
loop in which the plant secretes even more simple coumarins to compensate; these additional
coumarins are, in turn, degraded again by NyZ480, thereby progressively compromising plant
iron acquisition and overall health.” (Lines 558-563).

Such phrasing moderates our originally definitive tone and avoids presenting catabolism as
the primary driver.

3. while the second isolate is helpful, it is very closely related, and thus does not yet convincingly
address the broader generality of the proposed mechanism across more diverse rhizosphere bacteria.

We thank the reviewer for this insightful point. While we agree that demonstrating this
phenomenon across diverse rhizosphere bacteria would broaden the study's generality, we
believe the validation using the second isolate, NyZ490, sufficiently demonstrates that the
proposed mechanism is not an anomaly of a single isolate.

Regarding the phylogenetic proximity of NyZ480 and NyZ490, it is important to note that
*Pseudomonas* is a cosmopolitan and ecologically significant genus in soil (Lines 545-546). Our
results indicate that the ability to utilize and detoxify simple coumarins for colonization may
be a conserved trait among certain *Pseudomonas* species. Thus, it remains possible that this is
a common occurrence in natural environments where this genus is dominant.

Most importantly, this study primarily focuses on presenting a previously undescribed
bacterial mechanism underlying plant-microbe interaction. Whether this trait is unique to
*Pseudomonas* or shared by other taxa, the discovery that bacteria can exploit plant-secreted
simple coumarins to affect plant growth and modulate plant iron status represents a
significant advance in itself. Since simple coumarins are essential to the health of diverse plant
species, we believe that our findings offer meaningful implications for both plant nutrition and
immunity fields.

Overall, I think the authors have done a good job and the manuscript is clearly much stronger now.
My recommendation would be that the study is close to being acceptable, but still slightly weak
mechanistically. In particular, further clarification could come from experiments that more directly
demonstrate in planta coumarin catabolism, for example by tracking coumarin derived carbon into
bacterial biomass, by measuring specific breakdown products in the rhizosphere during colonization,
or by using bacterial mutants that separate detoxification capacity from growth on coumarins as a
carbon source. Alternatively, if such experiments are beyond the scope of the current revision, the
conclusions should be framed more cautiously to reflect that the current data support a strong
correlation and functional dependence, but not yet an unambiguous demonstration of a clear
mechanism of nutritional exploitation.

**We sincerely thank the reviewer for the encouraging assessment of our revised manuscript**
**and for acknowledging the improvements made. We deeply appreciate your rigorous**
**standards regarding the mechanistic evidence and the constructive experimental suggestions**
**you provided, such as tracking coumarin-derived carbon or using specific separation-of-**
**function mutants.**

**We fully agree that these experiments would provide a more direct demonstration of *in planta***
**catabolism. However, as you graciously noted, establishing these complex experimental**
**systems is technically challenging and perhaps falls beyond the scope of the current revision.**
**We will certainly consider employing these advanced approaches, such as stable isotope**
**probing, in our future follow-up studies to further dissect these mechanisms.**

**Therefore, we have adopted your alternative recommendation to adjust the framing of our**
**conclusions. In the revised manuscript, we have carefully modified the text to present our**
**findings more cautiously, removing claims that present bacterial catabolism unambiguously**
**as the sole driver of NyZ480 colonization and also introducing the possibility of a concurrent**
**detoxification. The revised text is provided above (Lines 98-165 in this response letter). We**
**hope these revisions satisfactorily address your concerns.**

**Reviewer #3 (Remarks to the Author):**

The authors have done an impressive amount of additional work which greatly strengthens the
manuscript and addresses the majority of my initial concerns. My most major comments were that
ICP-MS analysis of plant iron status and the use of coumarin null plants were needed to definitively
link coumarin degradation and iron sufficiency/deficiency. The authors have provided a nice
demonstration of changes in iron content and have used coumarin null plants to link their phenotypes
to coumarins. I also suggested that the authors use CFU counts instead of qPCR to test root
colonization, again they have provided the appropriate experiments.

**We thank the reviewer for the kind words and for recognizing the effort we put into the**
**revision. We are delighted to hear that the additional data have satisfactorily addressed your**
**major comments. We would like to emphasize that it is your constructive suggestions that**
**directly guide these improvements. These critical experiments have significantly strengthened**
**our evidence base and enhanced the credibility of our conclusions.**

I do have one remaining comment (below) about missing controls for microscopy experiment.
However, as these data are not crucial for the paper they could be moved to the supplement without
the need for further experiments. If the data are going to remain in the main text, it will be important
to address this comment. Beyond this my concerns can mostly be addressed with modification of
the text rather than additional experiments.

**We appreciate the reviewer's comment regarding the microscopy controls. We fully agree with**
**your assessment that the confocal microscopy images serve as supporting data, whereas the**
**CFU counts provide the primary and quantitative evidence for bacterial colonization.**
**Therefore, we have gladly accepted your suggestion to move the microscopy results from the**
**main text to the Supplementary Information (now Supplementary Fig. 5). We believe this**
**adjustment streamlines the main text while preserving the data for interested readers, without**
**necessitating further experimental controls at this stage.**

This first pertains to the question of catabolism and detoxification. The MS still emphasizes
catabolism, but the data suggest that detoxification is most likely at work, at least some of the time.
I agree with the authors that the differences in WT bacterial colonization between iron deficient and
iron sufficient Arabidopsis strongly supports the hypothesis of catabolism. However, this does not
fully rule out detoxification and an experiment directly demonstrating coumarin catabolism in planta
would be needed to prove this one way or the other. This type of experiment would be extremely
difficult, and I am not suggesting that the authors attempt this. Instead, I suggest that the possibility
of detoxification be introduced earlier in the manuscript, that some of the claims of catabolism be
qualified, and that they provide more discussion of the possibilities for detoxification and catabolism
in different circumstances.

**We sincerely thank the reviewer for reiterating the critical distinction and interplay between**

catabolism and detoxification of simple coumarins during NyZ480 colonization. We
acknowledge that this is a pivotal issue, which was also raised by Reviewer 2 during this round
of review.

We concur that while the data strongly suggest catabolism is a major driver, detoxification
cannot be excluded. As you rightly noted, experimentally separating these two functions is
difficult because they share the same initiating gene, *xenA*. Therefore, generating mutants to
decouple these pathways is not currently feasible. We are very grateful that the reviewer
recognizes the technical hurdles and have suggested textual modifications instead.

Following your advice, we have introduced the possibility of detoxification earlier in the
manuscript, tempered our claims regarding catabolism as the sole driver for NyZ480
colonization, and added a more balanced discussion on how both processes likely operate
under different physiological contexts.

The revisions are made as follows:

(3) In order to incorporate the contributory role of detoxification, revisions are made in
several parts of the manuscript:

“Hence, we reveal an unreported rhizospheric phenomenon in which microorganisms
opportunistically utilize and detoxify host-secreted specialized metabolites under stress
conditions, enhancing colonization and impairing plant fitness.” (Lines 29-32).

We changed “degrade and utilize” to “utilize and detoxify”

“Our findings establish that bacterial utilization and detoxification of plant-derived simple
coumarins drive enhanced bacterial colonization and worsened growth impairment in iron-
stressed *Arabidopsis*.” (Lines 90-93).

We changed “degradation and utilization” to “utilization and detoxification”.

“Given that these plant-derived compounds are widely present in the rhizosphere and exhibit
antimicrobial activity, the redundant *xenA* genes harbored by the rhizosphere-isolated
NyZ480 likely represent an adaptive detoxification strategy, evolved in the ongoing tug-of-war
between rhizospheric microorganisms and their host plants.” (Lines 183-187).

We changed “may be considered as genetic arsenals” to “likely represent an adaptive
detoxification strategy”, which would allow to introduce the concept of detoxification earlier
and better emphasize its potential functional significance.

“Having demonstrated, using the *Arabidopsis* mutant *fb'h1*, that host production of simple
coumarins is indispensable for the enhanced colonization of WT-NyZ480 under iron deficiency,
we employed NyZ480 mutants to further investigate whether the bacterial_utilization and
detoxification capacity against these phytocompounds underpinned its proliferation on iron-
stressed plant roots.” (Lines 316-320).

We changed “degradation and utilization” to “utilization and detoxification”.

**“Collectively, xenA redundancy equips NyZ480 with the dual capacity to utilize certain simple**
**coumarins as carbon sources and to detoxify others’ antimicrobial effects.” (Lines 348-350).**
**We changed “withstand antimicrobial effects of others” to “detoxify others’ antimicrobial**
**activity”.**

**(4) In order to qualify the definitive conclusion, the following sentences are revised:**

**“....., demonstrating that the disrupted plant iron acquisition was likely caused by WT-**
**NyZ480’s degradation of simple coumarins.”**

**We replaced “indeed” with “likely”, and “WT-NyZ480’ s degradation and utilization” with**
**“WT-NyZ480’ s degradation” (Lines 375-376). These changes qualify the conclusion and**
**thereby temper the inference regarding the primary role of coumarin catabolism (utilization).**

**“Since NyZ480 not only degrades but also grows on simple coumarins, this continual**
**degradation and utilization massively deplete the iron-mobilizing coumarins, likely further**
**compromising plant uptake of the limited iron sources.” (Lines 470-473).**

**We added “likely” to qualify the conclusion.**

**“This proposed framework explains why” (Line 477).**

**We added “proposed” to moderate the inferred conclusion.**

**“In particular, the iron deficiency-induced enhanced colonization of NyZ480 further perturbs**
**plant iron acquisition through continuous degradation of simple coumarins,” (Line 520-**
**522).**

**We changed “through the degradation and utilization of simple coumarins” to “through**
**continuous degradation of simple coumarins”, which reduces emphasis on catabolism**
**(utilization) as the main driver underlying NyZ480’s enhanced colonization and subsequent**
**negative impact on plants.**

**We changed “It is the sustained bacterial growth on simple coumarins that perpetuates the**
**elimination of these iron-mobilizing SMs constantly secreted by plants, thereby establishing a**
**“vicious circle” that progressively compromises plant health.” to “Here, we propose that the**
**sustained degradation of simple coumarins by NyZ480 perpetuates the elimination of these**
**iron-mobilizing SMs from the rhizosphere. This persistent removal may create a feedback**
**loop in which the plant secretes even more simple coumarins to compensate; these additional**
**coumarins are, in turn, degraded again by NyZ480, thereby progressively compromising plant**
**iron acquisition and overall health.” (Lines 558-563).**

**Such phrasing moderates our originally definitive tone and avoids presenting catabolism as**
**the primary driver.**

**Also, in general, while the trends observed in the paper are strong, the MS makes no mention of**
**how things may change in more complex microbial communities in the environment. While NyZ480**
**may grow robustly as a monoculture this might shake out very differently if other microbes were**

around, the authors should provide a small discussion of this complexity and contextualize some of
their claims accordingly.

**We sincerely thank the reviewer for raising this insightful point. We fully acknowledge that**
**our study characterizes the impact of NyZ480 on iron-deficient Arabidopsis primarily as a**
**monoculture under controlled laboratory conditions, which inevitably differs from the**
**dynamics within complex microbial communities in natural rhizospheric environments.**

**We agree that this distinction is critical for interpreting the broader ecological relevance of**
**our findings. Following your suggestion, we have added a discussion in the revised manuscript**
**(Lines 570 – 578) to address this complexity. We explicitly state that while our monoculture**
**experiments reveal a potential mechanism of plant-microbe interaction, the situation in soil**
**environments is likely more intricate due to competition and cooperation among diverse**
**microbes. We have contextualized our claims to reflect that our findings suggest a mechanism**
**that might operate in nature, but further research is needed to validate these interactions**
**within complex community settings.**

**The added discussion reads: “These findings underscore that specific microbes may subvert**
**plant’s adaptive mechanism into an exploitable vulnerability. Given that iron deficiency**
**challenges global agricultural productivity, greater attention should be directed to the**
**potential disruption of plant iron acquisition by such coumarin-degrading rhizobacteria. We**
**acknowledge that interactions in native soil microbiomes are likely more intricate due to**
**competition and cooperation among diverse microbes, the generalizability of this**
**monoculture-based mechanism requires further validation. Nonetheless, this study provides a**
**crucial proof-of-concept that such microbial exploitative strategy can occur.”**

Finally, the writing of the text is much improved over the previous version but still requires further
editing before it will be of publication quality. A few examples are pointed out below, each is minor
but as a whole, they detract from the MS.

Small comments:

Line 26: I think it is hard to say that the bacterium is “overproliferating” , I would focus this on
the results.

**We appreciate the reviewer’s insightful comment. We agree that the term “over-proliferate”**
**might be an overstatement. Accordingly, we have removed it in the revised manuscript (Line**
**26) to more accurately reflect our results.**

Line 32: “to enhance colonization and impair plant fitness” I think this is too strong. While I
agree that this is the outcome, the “purpose” from the bacterial standpoint is ambiguous. I suggest
something like “enhancing colonization and impairing plant fitness.”

**We appreciate the reviewer’s suggestion. We agree that the original phrasing implied intent,**

**whereas the suggested wording more accurately describes the biological outcome. We have**
**revised the text to “enhancing colonization and impairing plant fitness” as suggested (Line 32).**

Line 58: Into the rhizosphere

**We have revised the text accordingly.**

Line 64: “Although it still lacks explicit knowledge regarding the composition...” This sentence
is a bit confusing.

**We have changed the original sentence to “While the exact composition and quantity of simple**
**coumarins exuded by pathogen-infected plants remain to be fully characterized,**
**transcriptomic analyses have preliminarily indicated that foliar infection activates**
**biosynthesis of simple coumarins” (Lines 62-65). Hopefully, the revised expression provides**
**clarity.**

Line 68: Daphnetin, with an a

**We have revised the word accordingly.**

71-72: The logic is not clear here

**We thank the reviewer for pointing out the logical issue. We have rephrased the sentence to**
**explicitly link SM secretion to their mediating role in interactions.**

**Revised text: “Since SMs are secreted into the rhizosphere as root exudates, they likely**
**mediate sophisticated interactions between plants and rhizobacteria.” (Lines 70-71).**

97-98: “An intensive plant-microbe arms race has thus been revealed” . I found this a bit too
sweeping of a statement.

**We thank the reviewer for pointing this out. We agree that the original statement was too**
**definitive. We have revised the sentence to “a potential evolutionary “arms race” between**
**plants and rhizobacteria can be inferred” (Lines 96-97).**

Line 129: versatily eliminate? I am not sure what this means, this could be clarified.

**We thank the reviewer for pointing this out. The wording “versatily eliminate” was indeed**
**confusing. We have revised it into “efficiently degrade” (Line 128).**

Line 140: Compared to the control group

**We have made revision accordingly.**

Line 187: Arsenal implies multiple weapons. To me, this is just one among many.

**We agree with the reviewer. We have revised the sentence into “likely represent an adaptive**
**detoxification strategy.” (Line 185-186).**

Line 230: Induced the mildest influences = had the mildest effects?

**We thank the reviewer for this suggestion. Indeed, "had the mildest effects" is clearer and**
**more conventional. We have revised the text accordingly (Line 229).**

Line 285: On the plant’ s

**We have revised it accordingly.**

Line 286: Physiological levels, plural

**We have revised it accordingly.**

Line 287: Please cite PMID: 24246380

**We thank the reviewer for the careful reminder. We have cited the reference as requested.**

Line 294: grew limitedly = growth of this strain was limited?

**We appreciate the reviewer's attention to wording. Indeed, "limitedly" was not the most**
**precise term in this context. To more accurately convey that the bacteria exhibited only faint**
**growth on *f6'h1* roots, we have replaced it with "faintly" in the revised text.**

Line 301: Regarding?

**We acknowledge that the original phrasing here was somewhat unclear. We have revised the**

sentence to: “When examining the impact of WT-NyZ480 on *f6'h1* under different iron
conditions, it was found that.....” (Lines 300-301).

Line 360: More declined = decreased?

**We agree with the reviewer. We have replaced “more declined” with “decreased”.**

Line 377: While bacterial use of coumarins seems likely, there is no direct evidence for this so this
claim should be softened.

**We agree with the reviewer that the original claim was too strong given the lack of direct
evidence. We have softened the statement in the revised manuscript. The sentence now reads:
“....., demonstrating that the disrupted plant iron acquisition was likely caused by WT-
NyZ480’s degradation of simple coumarins” (Lines 375-376).**

Line 408-409: This sentence needs a bit of clarification

**We thank the reviewer for requesting clarification on this point. We have rewritten the
sentence to more explicitly link the enzymatic transformation to structural disruption and
subsequent detoxification. The revised text reads: “The presence of *xenA* in bacteria suggests
a capability for the initial transformation of simple coumarins. This breakdown of the
coumarin structure is likely sufficient to detoxify these antimicrobial plant-derived SMs,
thereby conferring bacterial resistance.” (Lines 407-409).**

Line 438: As above, I think it is hard to say that the bacterium is “overproliferating” , I would
focus this on the results rather than editorial comments about it. Also, this sentence is a bit confusing
and needs to be revised.

**We sincerely thank the reviewer for this valuable comment. We completely agree that the term
“over-proliferate” was subjective and inappropriate. We have removed all editorial or
interpretive wording and now describe the phenomenon strictly based on the experimental
results. The sentence has been revised as follows:**

**“In this study, we demonstrate that the rhizobacterium *Pseudomonas* sp. NyZ480 possesses
robust detoxification and catabolic capabilities for simple coumarins, a key class of plant
secondary metabolites synthesized and secreted to facilitate iron acquisition and pathogen
defense. This capacity enables NyZ480 to establish enhanced colonization on iron-deficient,
coumarin-secreting *Arabidopsis*, thereby severely impairing plant iron acquisition and overall
fitness.” (Lines 433-438).**

Lines 558-561: I don't think there is strong evidence to support this statement. To prove that this
is the case it would be necessary to demonstrate that bacteria are growing on coumarins rather than
other exuded carbon in planta, (which would be extremely difficult to do). However, without such
an experiment, it remains unclear whether bacteria are detoxifying or catabolizing coumarins.

**The reviewer is absolutely right that without direct *in planta* evidence, we cannot definitively**
**prove that NyZ480 is actually using the large amounts of simple coumarins secreted by iron-**
**deficient plants as the primary carbon sources responsible for the observed effects. We also**
**very much appreciate your understanding that obtaining such *in planta* evidence is technically**
**challenging.**

**Our original statement was indeed overly definitive and went beyond what the current data**
**can firmly support. Accordingly, we have toned down the wording considerably and now**
**present it more cautiously as a plausible and likely explanation rather than a proven fact. The**
**revised text now reads:**

**“Here, we propose that the sustained degradation of simple coumarins by NyZ480 perpetuates**
**the elimination of these iron-mobilizing SMs from the rhizosphere. This persistent removal**
**may create a feedback loop in which the plant secretes even more simple coumarins to**
**compensate; these additional coumarins are, in turn, degraded again by NyZ480, thereby**
**progressively compromising plant iron acquisition and overall health.” (Lines 558-563).**

**We believe this revised phrasing is much more balanced, appropriately speculative, and fully**
**respects the limitations of the current experimental evidence. We hope this meets with your**
**approval. Thank you again for helping us improve the rigor and accuracy of the manuscript.**

Figure comments:

Overall the figure captions need to provide more detail on the number of replicates (technical and
biological) as well as what the displayed error bars are (SD etc.)

**We sincerely thank the reviewer for this valuable suggestion. We have made the corresponding**
**revisions as requested at the following locations:**

**Main text: Lines 982 – 983, Lines 997 – 998, Line 1032-1033.**

**Supplementary Information: Lines 148 – 149, Lines 168 – 169.**

**We hope these changes fully address your concerns and are to your satisfaction.**

Fig 1B. Small comment: Perhaps just add +10 μ M + NyZ480 to the legend

**We thank the reviewer for the suggestion. Fig. 1b has been revised accordingly as requested.**

Fig.1D: Please state the number of replicates used and what the line and shaded region represent.

**We thank the reviewer for pointing this out. We have now added the following information to**
**the legend of Fig. 1d (Lines 990-992): “Solid lines represent the mean values of three biological**
**replicates (n=3), and shaded areas indicate the standard deviation (SD).”**

Fig. 3d: Add “control” or “untreated” or some other designation to indicate the no bacteria
treatments

**Thank you for this helpful suggestion. Following your recommendation, we have now labeled**
**the no-bacteria control as “mock” in Fig. 3c (previous Fig. 3d). For consistency, we have**
**applied the same “mock” designation to the corresponding no-bacteria controls in Fig. 3b&c,**
**Fig. 5b, and Supplementary Fig. S13d&e.**

Fig 3A/5A: The CFU counts are hard to see, please decrease the y-axis scale to better match the
data (mostly in 5A). Also, the statistics are a little confusing. How were the comparisons that are
displayed chosen? Please clarify. The box plots make it hard to track the differences between the
different conditions perhaps consider presenting these data as a line graph?

**Thank you for the suggestions.**

**Regarding the y-axis scale in Fig. 3a&5a, we have decreased the y-axis range in Fig. 5a so that**
**the CFU data are more clearly visible. For Fig. 3a, we decided to retain the original scale**
**because it allows intuitive visual comparison of the differences between the two groups, which**
**is a key point of that panel.**

**For the statistical comparisons, we performed the analyses as follows: within each treatment**
**(sFe and dFe separately), data of Day 0 were compared to that of Day 7 to determine whether**
**the bacterium significantly colonized the roots over the 7-day period; additionally, data of sFe**
**and dFe groups on Day 7 were statistically compared to test whether different iron conditions**
**resulted in significantly different bacterial colonization. This information has been explicitly**
**described in the revised figure legends. (Lines 1021-1022; Lines 1064-1065)**

**Concerning the presentation format, while we fully appreciate that connected line graphs can**
**effectively illustrate temporal trends, we respectfully prefer to retain the box plots. Our**
**rationale is as follows: The distinct colors already clearly separate the treatment groups, and**
**the side-by-side box plots allow immediate visual assessment of both within-day differences**
**across conditions and the overall data distribution on any given day, which we believe remains**
**highly intuitive for readers. We hope the reviewer finds this format acceptable.**

**Thank you again for the constructive comments that have helped improve the clarity of the**
**figures.**

Fig 3B/5B: These experiments require further controls. The authors have used different replicating
plasmids with RFP for the WT as opposed to the mutant strains. Were microscopy experiments done
using kanamycin/tetracycline/other antibiotics to maintain the plasmid in NyZ480? Otherwise, the
signal could be missed due to plasmid loss. In addition, the use of two different plasmids could lead
to differences in RFP expression levels. It would be nice to show that RFP is produced at equivalent
amounts and is not lost when antibiotics are absent within each strain and/or to show a positive
control for the mutant strains in 3B to ensure the difference is not just due to low fluorescence. In
addition, in 5B where the microscopy images show no colonization it is important to include a
positive control. These experiments are not strictly necessary for the paper as the CFU data provide
strong evidence, so if this cannot be improved the data could be moved to the supplement.

**We sincerely thank the reviewer for this very insightful and rigorous comment. Upon careful**
**consideration, we fully agree that the fluorescence microscopy experiments in Fig. 3b and Fig.**
**5b lack sufficient plasmid maintenance and expression controls, which indeed compromises**
**their rigor. As the reviewer also kindly pointed out, the CFU quantification already provides**
**clear evidence of bacterial colonization. Therefore, following the reviewer's suggestion, we**
**have moved these fluorescence microscopy panels from the main figures to the Supplementary**
**Materials (now presented as Supplementary Fig. 5), where they serve only as supportive visual**
**illustration rather than primary evidence. We believe this adjustment appropriately addresses**
**the concern while maintaining the robustness of the main conclusions. Thank you again for**
**this valuable recommendation.**

**Reviewer #4 (Remarks to the Author):**

I co-reviewed this manuscript with one of the reviewers who provided the listed reports. This is part
of the Nature Communications initiative to facilitate training in peer review and to provide
appropriate recognition for Early Career Researchers who co-review manuscripts.

**Many thanks.**

Point-by-point response letter
**Rhizobacteria opportunistically boost colonization and impair plant**
**fitness by degrading plant-derived coumarins under iron deficiency**

By Yichao Gu et al.

**Reviewer #2 (Remarks to the Author):**

The authors have addressed all my concerns and I am happy to recommend this manuscript for
publication.

**Many thanks.**

**Reviewer #3 (Remarks to the Author):**

The authors have adequately addressed my concerns and the manuscript is scientifically sound and
the claims are reasonable. My only remaining recommendation is that the authors conduct further
editing of writing for clarity and the removal of persistent typos and confusing phrasing.

**Thank you for your approval of our revised manuscript. We have thoroughly checked and**
**corrected the typos and ambiguous sentences throughout the paper as you suggested. All**
**changes are clearly marked in the Microsoft Word document using track changes mode.**

**Reviewer #4 (Remarks to the Author):**

I co-reviewed this manuscript with one of the reviewers who provided the listed reports. This is part
of the Nature Communications initiative to facilitate training in peer review and to provide
appropriate recognition for Early Career Researchers who co-review manuscripts.

**Many thanks.**